# A Prototype-oriented Fast Refinement Model for Few-shot Industrial Anomaly Detection

## Abstract

Industrial Anomaly Detection (IAD) in low data regime is crucial for automating industrial inspections in practice. Previous methods have primarily focused on obtaining robust prototypes using only a few normal images per product. However, these methods seldom account for transferring the characteristics of online query images to enhance the representativeness of the original prototypes in a systematic way. To address the pivot issue, we propose a prototype-oriented fast refinement model for few-shot IAD. Given online query images, we formulate prototype refinement as a nested optimization problem between transport probability for anomaly suppression and transform matrix for characteristic transfer. Then we present an Expectation Maximization (EM)-based algorithm to iteratively compute the transport probability and transform matrix. In the E-step, we use entropy-based optimal transport, known as the Sinkhorn algorithm, to learn the transport probability. In the M-step, the transform matrix is updated via gradient descent. Finally, we integrate our model with two popular and recently proposed few-shot IAD methods, PatchCore and WinCLIP. Comprehensive experiments on three widely used datasets including MVTec, ViSA, and MPDD verify the effectiveness and efficiency of our proposed model in few-shot IAD applications.

## 1 Introduction

Industrial Anomaly Detection (IAD) aims to automatically identify defects on product surfaces Liu et al. (2024) and has been attracting tremendous attention Zhao (2023); You et al. (2022); Lu et al. (2023). However, the fragmented nature of industrial anomalies ranging from subtle bruises to obvious breakages, with varying appearances and scales Roth et al. (2022) , making it difficult for fully-supervised methods to detect them He et al. (2017); Kamat & Sugandhi (2020). Therefore, unsupervised IAD methods, trained with massive normal product images, have been developed recently Liu et al. (2023); Lu et al. (2024). In practice, it is not always possible to obtain a large number of normal images for different products, making existing methods less effective due to their inability to generalize across products in low-data regime at test time Huang et al. (2022).

To tackle this challenge, few-shot learning Snell et al. (2017); Sung et al. (2018); Wang et al. (2020) has been introduced to unsupervised IAD, allowing the development of a common model shared across multiple products and generalizing to new products with only a few normal training images, such as 1-shot per product. This new paradiagm is known as few-shot (unsupervised) IAD Huang et al. (2022), and primarily involves prototype-oriented methods Fang et al. (2023); Jeong et al. (2023); Santos et al. (2023). At training time, these methods typically use the statistics of a few normal training (support) images to construct a set of normal prototypes, also known as a memory bank. During inference, anomaly scores are computed by measuring the differences between test (query) images and normal prototypes using various distance functions. Anomalies are detected by comparing the anomaly score against predefined thresholds. Especially, Fang et al. (2023) further employ statistics of query images at test time to refine prototypes quickly. However, we find that point-to-point regularization as in Fang et al. (2023) does significantly limits the ability to transfer characteristics from query images to prototypes. Additionally, meta learning based few-shot IAD methods Wu et al. (2021); Huang et al. (2022) have been introduced to achieve fast generalization, whose performance is verified to be far behind of prototype-oriented methods Xie et al. (2023).

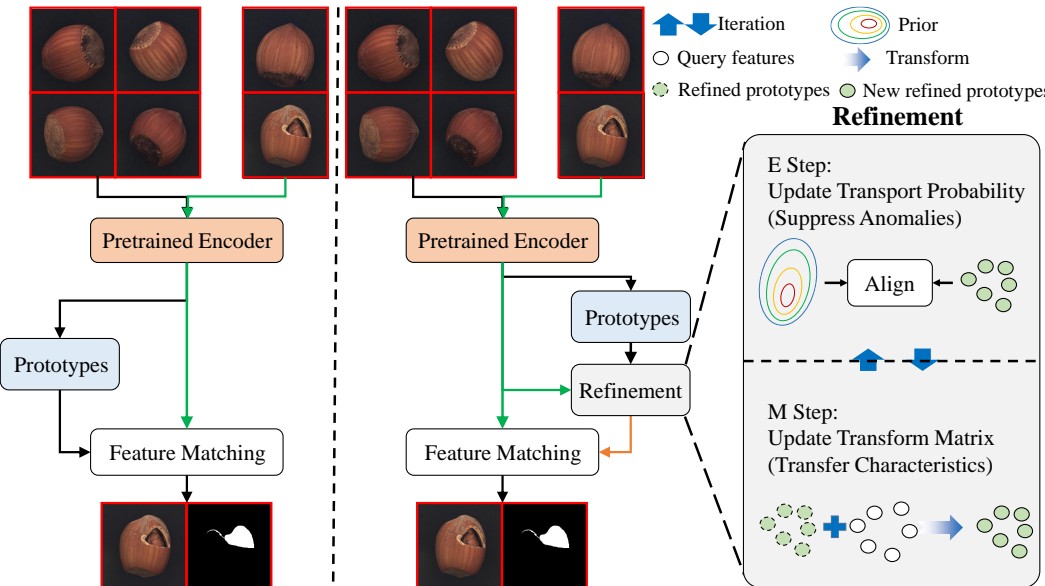

Figure 1: **Left:** Pipeline of prototype-oriented few-shot IAD methods. **Right:** Our proposed prototype-oriented fast refinement model for few-shot IAD. Our model leverages an EM-based optimization algorithm to refine prototypes by iterating between suppressing anomalies in query features and transferring characteristics from query features. In the E step, the refined prototypes are aligned with a prior distribution expanded using a few normal training features. In the M step, the refined prototypes are updated using both the old refined prototypes and query features.

Observing the fact that statistics of query images have not been fully explored at test time in the previous methods Fang et al. (2023); Jeong et al. (2023); Santos et al. (2023) either from data perspective Jeong et al. (2023); Santos et al. (2023) or optimization perspective Fang et al. (2023), which may result in suboptimal prototype refinement and cause the query features to deviate significantly from their normal prototypes, particularly when the available normal training images are extremely limited. To address this problem, we propose a prototype-oriented fast refinement model to transfer characteristics from query images to original prototypes while suppress anomalous features potentially present in query images, which results in more precise and efficient refinement, making the prototypes more generalizable. A comparison between our model and mainstream methods is shown in Fig. 1. Specifically, we frame prototype refinement with online query images as a nested optimization process balancing anomaly suppression and characteristic transfer, with a transport probability and transform matrix capturing these behaviors, respectively. We then introduce an EM-based algorithm to iteratively solve for the transport probability and transform matrix during inference. In the E-step, we use an entropy-regularized optimal transport to align the distribution between the original prototypes and the refined prototypes, ensuring the latter are unaffected by anomalies. In the M-step, gradient descent is employed to maximize the transfer of characteristics from query images to the refined prototypes. Finally, our model is integrated into two popular and recently proposed few-shot IAD methods, PatchCore Roth et al. (2022) and WinCLIP Jeong et al. (2023), to further enhance the representativeness of their prototypes. We find that our model consistently improves the performance of the two existing methods by significant margins across three widely used datasets including MVTec Bergmann et al. (2019), ViSA Zou et al. (2022), and MPDD Jezek et al. (2021). The main contributions could be summarized as follows:

- We present a prototype-oriented fast refinement model that explores the characteristics of query images and it can be integrated into existing methods like PatchCore and WinCLIP.

- We formulate prototype refinement as a nested optimization problem and introduce a novel EM-based algorithm to solve it precisely and efficiently at test time.

- The experimental results confirm that our proposed model is effective and significantly improves few-shot IAD performance on the MVTec, ViSA, and MPDD datasets.

## 2 RELATED WORKS

### 2.1 ANOMALY DETECTION

Industrial anomaly detection (IAD) involves handling training images that exclusively consist of normal data, primarily falling into two categories of reconstruction-based methods He et al. (2023); Wyatt et al. (2022); Gong et al. (2019); You et al. (2022); Lu et al. (2023) and memory-based methods Roth et al. (2022); Cohen & Hoshen (2020); Defard et al. (2021). Reconstruction-based methods are trained exclusively with normal images on the premise that anomalies will yield significantly higher reconstruction errors Gong et al. (2019). To address shortcut, Transformer-based architectures You et al. (2022); Lu et al. (2023) and diffusion-based training strategies Wyatt et al. (2022); Roth et al. (2022) have been developed concurrently. Memory-based methods take full advantages of pre-trained features to improve detection performance. However, both approaches tend to overfit when the number of normal training images per product is limited Huang et al. (2022).

### 2.2 FEW-SHOT ANOMALY DETECTION

Few-shot IAD has developed to address the demand for rapid manufacturing changeovers, with research mainly divided into prototype-oriented methods Santos et al. (2023); Xie et al. (2023); Fang et al. (2023); Jeong et al. (2023); Gu et al. (2024) and meta-learning based methods Wu et al. (2021); Huang et al. (2022). Prototype-oriented methods usually use pre-trained features to construct normal prototypes from only a few normal training images, with a focus on obtaining generalizable prototypes. Xie et al. (2023) develop graph Swin-Transformer Liu et al. (2021) to extract isometric-invariant visual features. Jeong et al. (2023); Li et al. (2024) turn to use Large Language Models (LLMs) Radford et al. (2021) to create powerful prototypes. Fang et al. (2023) leverage query image characteristics to enhance prototype representativeness. Although these methods efficiently build generalizable prototypes, they still lack a systematic way for refining prototypes properly.

## 3 BACKGROUND

### 3.1 TASK FORMULATION

We formally define the one-class IAD task in a low data regime, adhering to the standard few-shot learning. The model is fine-tuned using $k$ normal support images $\boldsymbol{x}_{1:k}^{\mathrm{s}}$ and predicts whether the $t$-th query image $\boldsymbol{x}_t^{\mathrm{q}}$ is anomalous at both the pixel and image levels. Notably, previous methods rarely utilize the statistics of query images $\boldsymbol{x}_t^{\mathrm{q}}$ for systematic predictions, often leading to suboptimal outcomes. In this paper, we fully leverage the pre-trained backbone $f_{\boldsymbol{\theta}^*}$, parameterized by $\boldsymbol{\theta}^*$, to extract features from both support and query images, computed as follows:

$$\boldsymbol{f}_{1:k \times h \times w}^{\mathrm{s}} = \mathrm{flatten}[f_{\boldsymbol{\theta}^*}(\boldsymbol{x}_{1:k}^{\mathrm{s}})], \quad \boldsymbol{f}_t^{\mathrm{q}} = f_{\boldsymbol{\theta}^*}(\boldsymbol{x}_t^{\mathrm{q}}) \tag{1}$$

where $\mathrm{flatten}[\cdot]$ is an operation that converts a 2-D feature map into a 1-D vector. Let $\boldsymbol{f}_l^{\mathrm{s}} \in \mathbb{R}^c$, where $l = 1, ..., k \times h \times w$, and $\boldsymbol{f}_t^{\mathrm{q}} \in \mathbb{R}^{h \times w \times c}$. In practice, normal features tend to be redundant, so compression techniques like Coreset Sener & Savarese (2017) are commonly employed to construct prototypes $\boldsymbol{\mathcal{M}}_{\mathrm{s}} \in \mathbb{R}^{\alpha \times k \times h \times w}$ by selecting the most representative normal features from $\boldsymbol{f}_{1:k \times h \times w}^{\mathrm{s}}$ with a downsampling ratio $\alpha \in (0, 1)$. For simplicity and conciseness, we denote $m = h \times w$ and $n = \alpha \times k \times h \times w$ for the reminder of this paper.

### 3.2 OPTIMAL TRANSPORT

Although Optimal Transport (OT) has a rich theory, we limit our discussion to OT for discrete distributions and refer the readers to Peyré et al. (2019) for more details. Let us consider $p$ and $q$ as two discrete probability distributions on the arbitrary space $\boldsymbol{X}, \boldsymbol{Y} \subset \mathbb{R}^d$, which can be formulated as $p = \sum_{i=1}^n a_i \delta_{\boldsymbol{x}_i}$, and $q = \sum_{j=1}^m b_j \delta_{\boldsymbol{y}_j}$. In this case, $\boldsymbol{a} \in \Sigma^n$ and $\boldsymbol{b} \in \Sigma^m$, where $\Sigma^n$ denotes the probability simplex of $\mathbb{R}^n$. The OT distance between $\boldsymbol{a}$ and $\boldsymbol{b}$ is defined as:

$$\mathrm{OT}(p, q) = \min_{\boldsymbol{T} \in U(p,q)} \langle \boldsymbol{T}, \boldsymbol{C} \rangle \tag{2}$$

where $\langle \cdot, \cdot \rangle$ denotes the Frobenius dot-product, $\boldsymbol{C} \in \mathbb{R}_{\geq 0}^{n \times m}$ is the transport cost function with element $C_{i,j} = \boldsymbol{C}(\boldsymbol{x}_i, \boldsymbol{y}_j)$, $\boldsymbol{T} \in \mathbb{R}_{>0}^{n \times m}$ is the doubly stochastic transport probability matrix such that

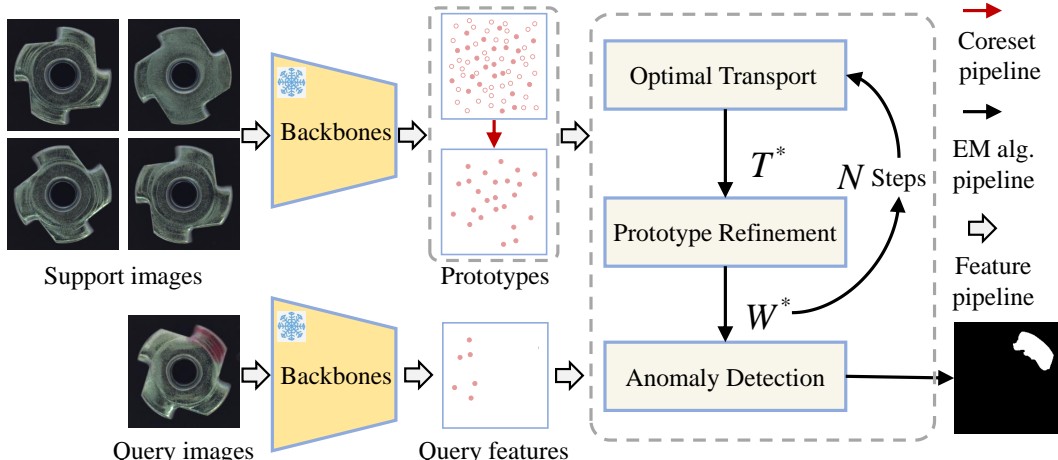

Figure 2: Overview of our proposed prototype-oriented fast refinement model. Support and query images are processed through a pre-trained backbone to extract normal and query features, respectively. Original prototypes are created by compressing normal features using Coreset Sener & Savarese (2017). We then employ EM-based optimization to refine prototypes by transferring characteristics while suppressing anomalies in the query features, where transport probability and transform matrix are iteratively updated until reaching the optimal values $\boldsymbol{T}^*$ and $\boldsymbol{W}^*$. Anomaly detection is performed by comparing the differences between query features and the refined prototypes. Notably, our model can be friendly integrated with other prototype-oriented methods, as discussed in Sec. 5.

$U(p, q) := \{\boldsymbol{T} \,|\, \sum_i^n \boldsymbol{T}_{i,j} = \boldsymbol{b}_j, \sum_j^m \boldsymbol{T}_{i,j} = \boldsymbol{a}_i\}$. To relax the time-consuming problem when optimizing the OT distance, Cuturi (2013) introduced the entropic regularization, $H = -\sum_{i,j} \boldsymbol{T}_{i,j} \ln \boldsymbol{T}_{i,j}$, leading to the widely-used Sinkhorn algorithm for discrete OT problems.

# 4 METHOD

We present our proposed model from the following perspectives: In Sec. 4.1, we formulate prototype refinement as a nested optimization problem. In Sec. 4.2, we introduce an EM algorithm to derive the refined prototypes. Finally, in Sec. 4.3, we implement anomaly detection through reconstruction using query images and the refined prototypes. An overview of our model is illustrated in Fig. 2.

## 4.1 A NESTED PROCESS FOR MODELLING PROTOTYPE REFINEMENT

We aim to improve the representativeness of prototypes by transferring characteristics while suppressing anomalies from query images. To achieve this, we propose a nested process to model these behaviors by:

$$\boldsymbol{W}^*, \boldsymbol{T}^* := \operatorname{argmin}_{\boldsymbol{W}, \boldsymbol{T}} \operatorname{dis}(\boldsymbol{f}_t^q, \boldsymbol{W}\boldsymbol{\mathcal{M}}_s) + \lambda \operatorname{OT}(p_s, q_s) \tag{3}$$

where $\operatorname{dis}(\cdot, \cdot)$ represents the distance between the two sets, with its form dependent on specific integrated methods discussed in Sec. 5. $\boldsymbol{W} \in \mathbb{R}^{m \times n}$ is the transform matrix used to transfer characteristics from the query features $\boldsymbol{f}_t^q$ to the original prototypes $\boldsymbol{\mathcal{M}}_s$, resulting in the refined prototypes $\widetilde{\boldsymbol{\mathcal{M}}}_s = \boldsymbol{W}\boldsymbol{\mathcal{M}}_s$, where $\widetilde{\boldsymbol{\mathcal{M}}}_s \in \mathbb{R}^{m \times c}$. $\boldsymbol{\mathcal{M}}_s$ and $\widetilde{\boldsymbol{\mathcal{M}}}_s$ are sampled from distributions of $p_s$ and $q_s$, respectively. Unlike the common assumption that refinement should occur along the feature dimension, in this work, it takes place along the sample dimension. We refer to the former as transform refinement and the latter as composition refinement. $\operatorname{OT}(\cdot, \cdot)$ represents the optimal transport distance described in Sec. 3.2 and can be viewed as a metric for suppressing anomalies potentially present in query images, as it ensures that the refined prototypes $\widetilde{\boldsymbol{\mathcal{M}}}_s$ are positioned in the high probability density region defined by the normal prototypes $\boldsymbol{\mathcal{M}}_s$. Importantly, our proposed regularization using the OT distance does not rely on a Gaussian distribution assumption, making it more adaptable for real-world few-shot IAD applications. $\lambda$ is the balanced coefficient. For optimization, we must further clarify the parsed formula for the OT distance in Eq. 3. Specifically,

given $\boldsymbol{\mathcal{M}}_s \sim p_s$ and $\widetilde{\boldsymbol{\mathcal{M}}}_s \sim q_s$, we can represent each feature in $p_s$ and $q_s$ as empirical distributions over the corresponding n and m features in their respective data spaces as follows:

$$p_s = \sum_{i=1}^{n} \frac{1}{n} \delta_{\boldsymbol{\mathcal{M}}_{s,i}}, \quad \boldsymbol{\mathcal{M}}_{s,i} \in \mathbb{R}^c; \quad q_s = \sum_{j=1}^{m} \frac{1}{m} \delta_{\widetilde{\boldsymbol{\mathcal{M}}}_{s,j}}, \quad \widetilde{\boldsymbol{\mathcal{M}}}_{s,j} \in \mathbb{R}^c \tag{4}$$

Moreover, we utilize an entropy-based OT distance Cuturi (2013) and formulate the optimization problem for the second term in Eq. 3 as follows:

$$\mathrm{OT}_\epsilon(p_s, q_s) := \sum_{i,j}^{n,m} \boldsymbol{C}_{i,j} \boldsymbol{T}_{i,j} - \epsilon \sum_{i,j}^{n,m} -\boldsymbol{T}_{i,j} \ln \boldsymbol{T}_{i,j} \tag{5}$$

where $\epsilon > 0$, $\boldsymbol{C} \in \mathbb{R}_{\geq 0}^{n \times m}$ is the cost matrix, which is typically formulated using simple distance functions $\mathrm{dis}(\cdot, \cdot)$, such as Euclidean or cosine. The specific form of $\boldsymbol{C}$ should align with the distance function used in the first term of Eq. 3, which we will discuss in Sec. 5. Importantly, the transport probability $\boldsymbol{T} \in \mathbb{R}_{>0}^{n \times m}$ must satisfy $U(p_s, q_s) := \{\sum_i^n \boldsymbol{T}_{i,j} = \frac{1}{m}, \sum_j^m \boldsymbol{T}_{i,j} = \frac{1}{n}\}$, where $\boldsymbol{T}_{i,j}$ denotes the transport probability between the i-th prototypes and the j-th refined prototypes, serving as an upper-bounded positive metric. Consequently, $\boldsymbol{T}_{i,j}$ naturally weights the importance of each refined prototype in relation the set of original (normal) prototypes.

### 4.2 AN EM ALGORITHM FOR SOLVING PROTOTYPE REFINEMENT

Observing the nested optimization problem in Eq. 3, it is clear that the optimal parameters for the transport probability $\boldsymbol{T}^*$ and the transform matrix $\boldsymbol{W}^*$ are interdependent. This motivates us to leverage the EM algorithm for iterative solving, as illustrated in Fig. 2. For the $t$-th iteration, in the E-step, we keep the transform matrix $\boldsymbol{W}_t$ fixed and update the transport probability $\boldsymbol{T}_t$ by minimizing the second term of Eq. 3 using Sinkhorn algorithm Cuturi (2013) to derive $\boldsymbol{T}_{t+1}$. In the M-step, we keep the transport probability $\boldsymbol{T}_{t+1}$ fixed and update transform matrix $\boldsymbol{W}_t$ by minimizing Eq. 3, denoted as $\mathcal{L}(\boldsymbol{f}_t^q, \boldsymbol{\mathcal{M}}_s; \boldsymbol{W}, \boldsymbol{T})$, using gradient descent as $\boldsymbol{W}_{t+1} = \boldsymbol{W}_t + \beta \frac{\partial \mathcal{L}(\boldsymbol{f}_t^q, \boldsymbol{\mathcal{M}}_s; \boldsymbol{W}_t, \boldsymbol{T}_{t+1})}{\partial \boldsymbol{W}_t}$. After N steps, the optimal refined prototypes can be expressed as $\widetilde{\boldsymbol{\mathcal{M}}}_s^* = \boldsymbol{W}^* \boldsymbol{\mathcal{M}}_s$, where $\boldsymbol{W}^* = \boldsymbol{W}_N$ and $\boldsymbol{T}^* = \boldsymbol{T}_N$. We find that N = 10 yields promising anomaly detection results, indicating that the optimization of our model is efficient. Additionally, we initialize the transform matrix with $\boldsymbol{W}_0 = (\boldsymbol{f}_t^q \boldsymbol{\mathcal{M}}_s^T)(\boldsymbol{\mathcal{M}}_s^T \boldsymbol{\mathcal{M}}_s)^{-1}$ for fast convergence.

### 4.3 ANOMALY DETECTION WITH RECONSTRUCTION

Once the optimal refined prototypes $\widetilde{\boldsymbol{\mathcal{M}}}_s^*$ are obtained, the anomaly score map $\boldsymbol{s}$ for the t-th query image $\boldsymbol{x}_t^q$ can be defined by calculating the similarities between $\widetilde{\boldsymbol{\mathcal{M}}}_s^*$ and $\boldsymbol{x}_t^q$ as follows:

$$\boldsymbol{s}_j := \min_{\boldsymbol{r} \in \boldsymbol{\mathcal{M}}_s^*} \mathrm{dis}(\boldsymbol{f}_{t,j}^q, \boldsymbol{r}), \quad j = 1, ..., m \tag{6}$$

For image-level anomaly detection, we represent the maximum score $s^*$ among all values in $\boldsymbol{s} \in \mathbb{R}^m$ as $s^* = \max_{j \in [1,m]} \boldsymbol{s}_j$. For pixel-level localization, we first upscale the anomaly score map $\boldsymbol{s}$ using bi-linear interpolation to match the original input resolution. Next, we smooth the score map with a Gaussian kernel of fixed width equal to 4, rather than optimizing this parameter.

## 5 INTEGRATION WITH PROTOTYPE-ORIENTED FEW-SHOT IAD METHODS

### 5.1 PATCHCORE+: INTEGRATION WITH PATCHCORE

PatchCore Roth et al. (2022) is originally introduced for one-class IAD with a large number of normal images. Recently, however, it has also been applied to few-shot IAD tasks, thanks to its flexible prototype-oriented design Santos et al. (2023). A notable drawback of PatchCore is that its prototypes remain fixed during inference, which causes it to overlook statistics in query images. To address this issue, we propose PatchCore+ by incorporating our prototype refinement model to enhance the representativeness of the original prototypes in PatchCore by effectively exploring the characteristics

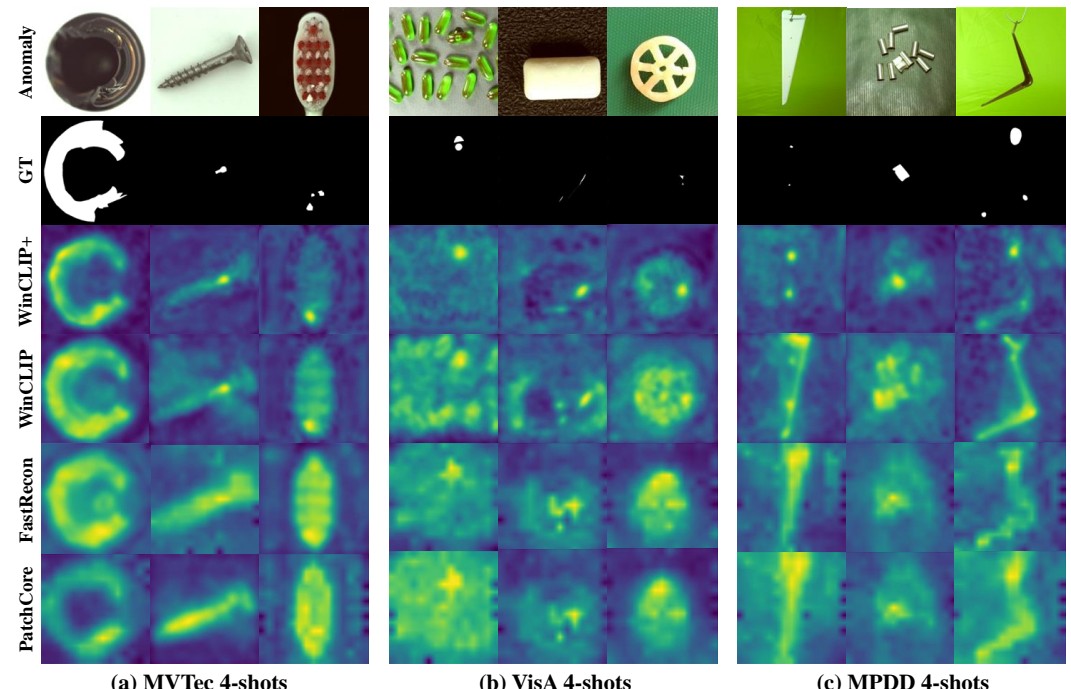

Figure 3: Qualitative results of pixel-level anomaly localization under 4-shots.

of query images. We use Euclidean distance in line with PatchCore and substitute it into Eq. 3 and Eq. 6, allowing us to rewrite the nested optimization problem and anomaly score map as follows:

$$\boldsymbol{W}^*, \boldsymbol{T}^* := \mathrm{argmin}_{\boldsymbol{W}, \boldsymbol{T}} \|\boldsymbol{f}_\mathrm{t}^\mathrm{q} - \boldsymbol{W}\boldsymbol{\mathcal{M}}_\mathrm{s}\|^2 + \lambda \mathrm{OT}(p_\mathrm{s}, q_\mathrm{s})$$

$$\boldsymbol{s}_\mathrm{j} := \min_{\boldsymbol{r} \in \boldsymbol{\mathcal{M}}_\mathrm{s}^*} \|\boldsymbol{f}_\mathrm{t,j}^\mathrm{q} - \boldsymbol{W}^*\boldsymbol{r}\|^2, \quad \mathrm{j} = 1, ..., \mathrm{m} \tag{7}$$

Once we have anomaly score map $\boldsymbol{s}$, anomaly detection can be carried out as described in Sec. 4.3.

## 5.2 WinCLIP+: Integration with WinCLIP

Unlike PatchCore whose support and query features are extracted from ResNet Zagoruyko & Komodakis (2016) or Efficient Tan (2019), WinCLIP Jeong et al. (2023) introduces a fine-tuning free large visual-language model based on the pre-trained CLIP model Radford et al. (2021). Although it demonstrates superior performance in few-shot IAD applications, it neglects the importance of efficiently transferring statistics from query images. To address this, we propose WinCLIP+ by integrating our prototype refinement model as a plug-and-play extension to WinCLIP, using consine distance to reformulate Eq. 3 and Eq. 6 as follows:

$$\boldsymbol{W}^*, \boldsymbol{T}^* := \mathrm{argmin}_{\boldsymbol{W}, \boldsymbol{T}} \frac{1}{2}[1 - \cos(\boldsymbol{f}_\mathrm{t}^\mathrm{q}, \boldsymbol{W}\boldsymbol{\mathcal{M}}_\mathrm{s})] + \lambda \mathrm{OT}(p_\mathrm{s}, q_\mathrm{s})$$

$$\boldsymbol{s}_\mathrm{j} := \min_{\boldsymbol{r} \in \boldsymbol{\mathcal{M}}_\mathrm{s}^*} \frac{1}{2}[1 - \cos(\boldsymbol{f}_\mathrm{t,j}^\mathrm{q}, \boldsymbol{r})], \quad \mathrm{j} = 1, ..., \mathrm{m} \tag{8}$$

To conduct anomaly detection, we combine the maximum value of $\boldsymbol{s}$ with the WinCLIP zero-shot anomaly score $s_0 : \mathbb{R}^\mathrm{c} \to [0, 1]$ for query features $\boldsymbol{f}_\mathrm{t}^\mathrm{q}$. These two scores provide complementary information, one from few-shot visual references and the other from CLIP knowledge by:

$$s^* = \frac{1}{2}[s_0(\boldsymbol{f}_\mathrm{t}^\mathrm{q}) + \max_{j \in [1, \mathrm{m}]} \boldsymbol{s}_j] \tag{9}$$

## 6 Experiments

We conduct comprehensive experiments on our proposed Patchcore+ and WinCLIP+ under 1-shot, 2-shots and 4-shots. We evaluate both image-level and pixel-level performances to demonstrate the

Table 1: Few-shot IAD performance averaged across on each dataset of MVTec, VisA and MPDD. Results of image-level and pixel-level are reported in AUROC/F1-max. The best results are in bold.

| Setup | Method | MVTec | | VisA | | MPDD | |
|---|---|---|---|---|---|---|---|
| | | Image | Pixel | Image | Pixel | Image | Pixel |
| 1-shot | PaDiM (ICPR'21) | 76.6/88.2 | 89.3/40.3 | 62.8/75.3 | 89.9/17.4 | 57.5/73.4 | 73.9/12.0 |
| | RegAD (ECCV'22) | 82.9/89.6 | 92.5/46.5 | —/— | —/— | 60.9/74.6 | 92.6/13.4 |
| | FastRecon (ICCV'23) | 85.7/91.2 | 93.2/48.6 | 76.2/78.8 | 96.7/36.7 | 74.1/83.9 | 96.3/25.3 |
| | PromptAD (CVPR'24) | 92.9/ — | 95.1/ — | **86.5**/ — | 96.2/ — | —/ — | —/ — |
| | PatchCore (CVPR'22) | 84.1/91.1 | 92.3/44.1 | 71.0/76.0 | 96.1/32.6 | 71.0/79.5 | 96.3/22.6 |
| | PatchCore+ (Ours) | 85.9/92.0 | 93.7/**50.4** | 78.3/79.6 | **97.1/37.6** | **74.9/84.3** | 96.6/26.0 |
| | WinCLIP (CVPR'23) | 93.5/93.7 | 93.6/43.2 | 83.4/81.9 | 94.7/22.9 | 70.5/81.2 | 96.3/31.4 |
| | WinCLIP+ (Ours) | **93.8/94.0** | **95.7**/48.2 | 83.9/**82.4** | 95.8/25.3 | 72.5/82.6 | **96.9/31.6** |
| 2-shot | PaDiM (ICPR'21) | 78.9/89.2 | 91.3/43.7 | 67.4/75.7 | 92.0/21.1 | 58.0/74.3 | 75.4/14.0 |
| | RegAD (ECCV'22) | 85.7/91.5 | 94.6/49.9 | —/— | —/— | 63.4/76.8 | 93.2/16.8 |
| | FastRecon (ICCV'23) | 88.3/92.5 | 94.5/51.9 | 86.1/82.3 | 97.6/42.8 | 76.4/83.8 | 96.7/29.5 |
| | PromptAD (CVPR'24) | 93.4/ — | 95.4/ — | 86.7/ — | 96.5/ — | —/ — | —/ — |
| | PatchCore (CVPR'22) | 87.1/92.2 | 93.3/46.4 | 80.0/79.1 | 96.9/36.8 | 71.4/80.7 | 96.5/24.8 |
| | PatchCore+ (Ours) | 88.8/93.4 | 94.7/**52.5** | **87.1/83.0** | **98.0/43.0** | **78.2/85.4** | 96.9/31.5 |
| | WinCLIP (CVPR'23) | 93.7/94.5 | 93.8/43.8 | 83.8/82.3 | 95.1/23.9 | 72.5/82.1 | 96.5/33.2 |
| | WinCLIP+ (Ours) | **93.9/94.8** | **96.2**/49.9 | 84.1/82.9 | 96.4/26.9 | 76.0/83.3 | **97.3/34.4** |
| 4-shot | PaDiM (ICPR'21) | 80.4/90.2 | 92.6/46.1 | 72.8/78.0 | 93.2/24.6 | 58.3/75.3 | 75.9/16.0 |
| | RegAD (ECCV'22) | 88.2/92.3 | 95.8/51.7 | —/— | —/— | 68.3/79.5 | 93.9/24.3 |
| | FastRecon (ICCV'23) | 91.3/93.8 | 96.1/53.8 | 88.2/83.1 | 98.0/44.6 | 79.7/85.9 | 95.2/33.7 |
| | PromptAD (CVPR'24) | 95.5/ — | 96.3/ — | 88.8/ — | 96.8/ — | —/ — | —/ — |
| | PatchCore (CVPR'22) | 90.0/93.4 | 95.1/49.9 | 84.2/80.7 | 97.5/38.1 | 76.2/84.1 | 97.2/28.5 |
| | PatchCore+ (Ours) | 92.1/94.4 | 96.1/**54.1** | **90.4/85.4** | **98.2/45.0** | 80.3/**87.4** | 97.2/**35.7** |
| | WinCLIP (CVPR'23) | 95.3/94.9 | 94.2/45.9 | 84.1/82.5 | 95.4/25.3 | 75.0/83.4 | 96.8/34.8 |
| | WinCLIP+ (Ours) | **95.5/95.1** | **96.7**/53.2 | 85.0/83.0 | 96.6/28.4 | **82.0**/84.1 | **97.6**/35.0 |

effectiveness of the our proposed prototype refinement model in few-shot IAD. Ablation studies are performed to validate the improvements brought about by the characteristics transfer and anomalies suppression. Finally, we analyzed the impact of various hyperparameters.

## 6.1 EXPERIMENT SETUP

**Datasets.** We conduct experiments on MVTec Bergmann et al. (2019), VisA Zou et al. (2022), and MPDD Jezek et al. (2021) datasets. The MVTec dataset consists of 3,629 training images and 1,725 test images across 15 categories, covering 5 types of textures and 10 types of objects, with each category exhibiting an average of five distinct defect types. Image resolutions ranging from 700×700 to 1,024×1,024. The VisA dataset contains 9,621 normal images and 1,200 anomaly images featuring 78 types of anomalies. It is divided into 12 subsets, each representing a distinct object, with an average of 6.5 defect types per subset. Image resolutions are around 1,500×1,000. The MPDD dataset includes 888 normal training images and 458 test images, including 176 normal and 282 abnormal images, spanning 6 classes of metal products with a resolution of 1,024×1,024.

**Competing Methods.** We compare our model against several recently proposed few-shot IAD methods or those applicable in low-data regimes, including PaDiM Defard et al. (2021), RegAD Huang et al. (2022), PatchCore Roth et al. (2022), FastRecon Fang et al. (2023), WinCLIP Jeong et al. (2023), and PromptAD Li et al. (2024). PaDiM, PatchCore and FastRecon are CNN-based methods. RegAD is meta-learning based model. WinCLIP and PromptAD are CLIP-driven methods. For fairness, we use the performance of PatchCore, FastRecon, WinCLIP and PromptAD using the same support images and official or reproduced code implementations.

**Evaluation Protocols.** Following previous methods Jeong et al. (2023), we evaluate the performance of anomaly detection and localization using image/pixel-level AUROC and image/pixel-level F1-max. Additionally, we assess real-time efficiency by measuring the running time per image.

**Implementation Details.** For PatchCore+, we use a pre-trained WRN-50 Zagoruyko & Komodakis (2016) to extract features from intermediate layers, following Roth et al. (2022). All images from the MVTec-AD, VisA, and MPDD datasets are resized to 256×256. Balanced coefficient $\lambda = 0.3$ and

Table 2: Ablation studies of WinCLIP+ with image-level and pixel-level AUROCs under 2-shots. The best results are in bold.

| $W^*$ | $T^*$ | MVTec | | VisA | | MPDD | |
|---|---|---|---|---|---|---|---|
| | | Image | Pixel | Image | Pixel | Image | Pixel |
| × | × | 93.1 | 93.8 | 83.4 | 95.1 | 72.5 | 96.2 |
| ✓ | × | 93.7 | 94.7 | 83.7 | 96.2 | 74.9 | 96.4 |
| ✓ | ✓ | **93.9** | **96.2** | **84.1** | **96.4** | **76.0** | **97.3** |

Table 3: Comparisons of inference time in seconds on MVTec.

| Methods | Inference time |
|---|---|
| PromptAD | 3.54 |
| PatchCore | 0.21 |
| PatchCore+ | 0.49 |
| WinCLIP | 0.50 |
| WinCLIP+ | 0.81 |

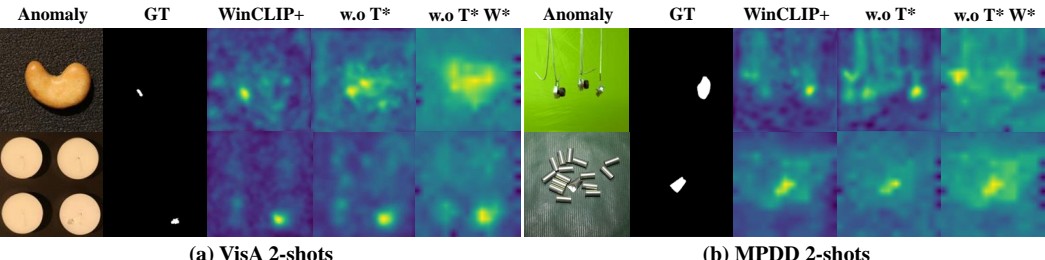

(a) VisA 2-shots          (b) MPDD 2-shots

Figure 4: Qualitative ablation studies of 2-shots anomaly localization on MVTec and VisA datasets.

Coreset sampling ratio $\alpha = 0.05$. For WinCLIP+, we set the image resolution to 240×240 and use the pre-trained CLIP model with ViT-B/16+ to extract image features for anomaly detection, following Jeong et al. (2023). In this case, $\lambda = 0.1$ and $\alpha = 0.5, 0.3, 0.2$ for MVTec-AD, VisA, and MPDD datasets, respectively. All experiments are conducted on a single NVIDIA GTX 3090 GPU.

## 6.2 COMPARISONS WITH SOTA METHODS

**Image-level Comparison Results.** We present image-level anomaly detection results in Table 4. PaDiM Defard et al. (2021) and PatchCore Santos et al. (2023) are adapted from traditional full-shot methods to few-shot settings. Comparing the results of RegAD Huang et al. (2022), PatchCore Santos et al. (2023), and FastRecon Fang et al. (2023), the following observations are evident: i) Prototype-oriented methods outperform the meta-learning based model, demonstrating the superior flexibility and generalizability of their feature representations; ii) FastRecon significantly outperforms Patch-Core, highlighting the importance of incorporating statistics from query images. Notably, PatchCore and FastRecon use CNN-based pre-trained features for few-shot IAD. When comparing PatchCore and FasRecon to WinCLIP Jeong et al. (2023), WinCLIP achieves a substantial performance gain except on the MPDD dataset. We attribute this to two factors: i) Generally, the representations from pre-trained CLIP models are more powerful than those from CNNs; ii) Unlike MVTec and VisA, MPDD is a metal dataset with rotation variations that may not be well-represented during CLIP pre-training. This discrepancy motivates us to demonstrate the effectiveness of our model using both CNN-based and CLIP-based pre-trained models. By comparing the results of PatchCore/WinCLIP with PatchCore+/WinCLIP+, we observe that PatchCore+/WinCLIP+ consistently delivers superior IAD performance, indicating that our prototype refinement model effectively addresses the challenges in few-shot IAD. For example, WinCLIP+ achieves a 7% improvement in AUROC on the MPDD dataset under 4-shots. Furthermore, the improvement delivered by our model surpasses that of the point-to-point regularization approach used in FastRecon Fang et al. (2023), underscoring the importance of refining prototypes in a more systematic way.

**Pixel-level Comparison Results.** Pixel-level anomaly localization results are presented in Table 4. When comparing PatchCore+ and WinCLIP+ with other competitive methods, we observe that the trends in pixel-level AUROC and F1-max are consistent with the image-level results. This consistency suggests that our model not only effectively detects anomalous images but also accurately localizes the anomalous regions. Once again, we attribute these performance gains to our well-designed prototype refinement model.

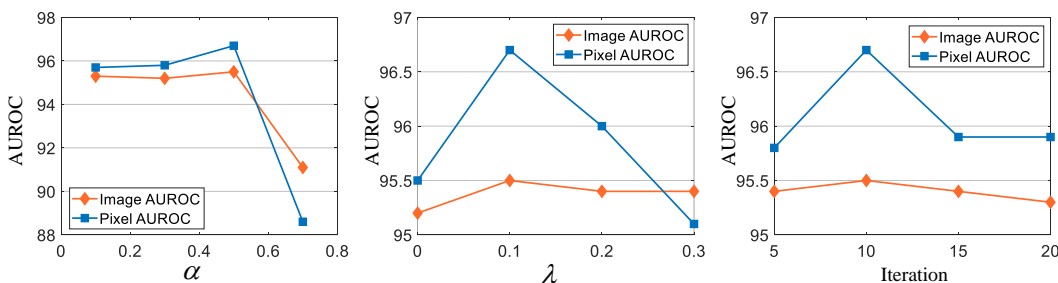

Figure 5: Hyper-parameters analysis on MVTec dataset under 4-shots.

**Qualitative results.** Visualization results for all three datasets are shown in Fig. 3. As demonstrated, our model (WinCLIP+) achieves more precise anomaly localization, particularly for subtle anomalies that are challenging to detect. This further validate the effectiveness of our approach.

## 6.3 ABLATIVE ANALYSIS

We utilize WinCLIP+ to assess the impact of each module in our prototype-oriented fast refinement model, including anomaly suppression via the optimal transport probability $T^*$ and characteristic transfer through the optimal transform matrix $W^*$ under 2-shots on all three datasets. The results are reported in Table 2, with visualizations for the MVTec and VisA datasets shown in Fig. 4.

**Impact of Anomaly Suppression by $T^*$.** Anomalies in query images negatively affect robust anomaly detection due to the limited diversity and representativeness of normal prototypes. Therefore, leveraging query images through effective anomaly suppression is crucial. As shown in Table 2, there is a notable performance decline in image/pixel level results when $T^*$ is not applied. Specifically, anomaly suppression yields improvements of over $1\%$ on both the MVTec and MPDD datasets.

**Impact of Characteristic Transfer $W^*$.** According to Table 2, $W^*$ consistently improves detection and localization performance across all three datasets. However, in same cases we observe that the gains from using $T^*$ are more pronounced than those from $W^*$. For example, in the case of pixel-level localization on the MPDD dataset, the gain from using $W^*$ is $0.2\%$, whereas further using $T^*$ results in a $0.9\%$ improvement. This suggests that relying solely on $W^*$ may introduce additional anomalies, thereby limiting overall performance improvements.

## 6.4 REAL-TIME EFFICIENCY

The running time per image during testing is compared in Table 3. PromptAD is the most time-consuming due to its complex prompt learning process. Overall, our proposed prototype refinement model adds only 0.3 s compared to its base models, PatchCore and WinCLIP.

## 6.5 HYPER-PARAMETERS ANALYSIS

In Fig. 8, we present results of the hyperparameters' impact under 4-shots on MVTec dataset , including the Coreset sampling ratio $\alpha$, the balanced coefficient $\lambda$, and the number of iteration N.

**Impact of CoreSet sampling ratio $\alpha$.** We observe that the Coreset sampling ratio $\alpha$ is crucial in determining IAD performance, as it controls the initial representativeness of the prototypes.

**Impact of the balanced coefficient $\lambda$.** We observe a similar trend as in $\alpha$ that the performance first improves and then declines as $\lambda$ changes. This phenomenon suggests that characteristic transfer should dominate the success of prototype refinement, aligning with our design.

**Impact of the iteration number $N$.** To obtain optimal refined prototypes, we need to update transport probability $T$ and transform matrix $W$ iteratively following an EM-based algorithm described in Sec. 4.2. Naturally, the iteration number N is crucial for robust few-shot IAD. Results reported in Fig. 8 (c) indicate that the WinCLIP+, enhanced by our proposed prototype refinement model, achieves strong IAD performance with N = 10, demonstrating that our model is efficient in practice. This also echoes the real-time efficiency discussed in Sec. 6.4.

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

# A  APPENDIX

Table 4: Few-shot IAD performance averaged across on each dataset of MVTec, VisA, MPDD, and RealIAD. Results of image-level and pixel-level are reported in AUROC. The values in parentheses represent the improvements of our method compared to the original method. The best and the second best results are bold with black and blue, respectively.

| Setup | Method | MVTec | | VisA | | MPDD | | RealIAD | |
|-------|--------|-------|-------|------|-------|------|-------|---------|-------|
| | | Image | Pixel | Image | Pixel | Image | Pixel | Image | Pixel |
| 1-shot | GraphCore (ICLR'23) | 89.9 | 95.6 | — | — | 84.7 | 95.2 | — | — |
| | PatchCore (CVPR'22) | 84.1 | 92.3 | 71.0 | 96.1 | 71.0 | 96.3 | 71.2 | 95.7 |
| | PatchCore+ (Ours) | 85.9(+1.8) | 93.7(+1.4) | 78.3(+7.3) | 97.1(+1) | 74.9(+3.9) | 96.6(+0.3) | 75.9(+4.7) | 96.3(+0.6) |
| | WinCLIP (CVPR'23) | 93.5 | 93.6 | 83.4 | 94.7 | 70.5 | 96.3 | 73.8 | 94.3 |
| | WinCLIP+ (Ours) | 93.8(+0.3) | 95.7(+2.1) | 83.9(+0.5) | 95.8(+1.1) | 72.5(+2.0) | 96.9(+0.6) | 74.4(+0.6) | 94.8(+0.5) |
| 2-shot | GraphCore (ICLR'23) | 91.9 | 96.9 | — | — | 85.4 | 95.4 | — | — |
| | PatchCore (CVPR'22) | 87.1 | 93.3 | 80.0 | 96.9 | 71.4 | 96.5 | 72.5 | 95.9 |
| | PatchCore+ (Ours) | 88.8(+1.7) | 94.7(+1.4) | 87.1(+7.1) | 98.0(+1.1) | 78.2(+6.8) | 96.9(+0.4) | 76.9(+4.4) | 96.5(+0.6) |
| | WinCLIP (CVPR'23) | 93.7 | 93.8 | 83.8 | 95.1 | 72.5 | 96.5 | 75.0 | 94.6 |
| | WinCLIP+ (Ours) | 93.9(+0.2) | 96.2(+2.4) | 84.1(+0.3) | 96.4(+1.3) | 76.0(+3.5) | 97.3(+0.8) | 75.9(+0.9) | 95.2(+0.6) |
| 4-shot | GraphCore (ICLR'23) | 92.9 | 97.4 | — | — | 85.7 | 95.7 | — | — |
| | PatchCore (CVPR'22) | 90.0 | 95.1 | 84.2 | 97.5 | 76.2 | 97.2 | 73.2 | 96.0 |
| | PatchCore+ (Ours) | 92.1(+2.1) | 96.1(+1) | 90.4(+6.2) | 98.2(+0.7) | 80.3(+4.1) | 97.2(+0) | 77.4(+4.2) | 96.7(+0.7) |
| | WinCLIP (CVPR'23) | 95.3 | 94.2 | 84.1 | 95.4 | 75.0 | 96.8 | 76.4 | 94.8 |
| | WinCLIP+ (Ours) | 95.5(+0.2) | 96.7(+2.5) | 85.0(+0.9) | 96.6(+1.2) | 82.0(+7) | 97.6(+0.8) | 77.3(+0.9) | 95.3(+0.5) |

Table 5: Ablation studies of WinCLIP+ with AUROC under 2-shot. The best results are in bold.

| $W^*$ | $T^*$ | MVTec | | VisA | | MPDD | |
|-------|-------|-------|-------|------|-------|------|-------|
| | | Image | Pixel | Image | Pixel | Image | Pixel |
| × | × | 93.7 | 93.8 | 83.8 | 95.1 | 72.5 | 96.5 |
| ✓ | × | 93.7 | 94.7 | 83.7 | 96.2 | 74.9 | 96.4 |
| ✓ | ✓ | 93.9 | 96.2 | 84.1 | 96.4 | 76.0 | 97.3 |

**Algorithm 1** Inference Process

**Require:** Initial transform matrix $W_0$, original prototypes $\mathcal{M}_s$, the $t$-th query features $f_t^q$
1: Calculating $p_s$ using $\mathcal{M}_s$
2: **For** $m = 0$ to $M - 1$ **do**
3:     Calculate $q_s$ using $W_m \mathcal{M}_s$
4:     **For** $e = 0$ to $E - 1$ **do**
5:         Update $T_{m+1}$ from $T_m$ by minimizing
6:         $OT(p_s, q_s)$ while fix $W_m$
7:         Update $W_{m+1}$ from $W_m$ by minimizing
8:         $\mathcal{L}(f_t^q, \mathcal{M}_s; W, T)$ while fix $T_{m+1}$
9: Set $W^* = W_M$, $\mathcal{M}_s^* = W^* \mathcal{M}_s$
10: Implement few-shot IAD as follows:
11: $s_j := \min_{r \in \mathcal{M}_s^*} \mathrm{dis}(f_{t,j}^q, r)$,    $j = 1, ..., m$

Table 6: Results of incorporating refined prototypes into the memory bank on MPDD.

| Method | 1-shot | | 2-shot | |
|--------|--------|-------|--------|-------|
| | Image | Pixel | Image | Pixel |
| PatchCore | 71.0 | 96.3 | 71.4 | 96.5 |
| PatchCore+ | 74.9 | 96.6 | 78.2 | 96.9 |
| Online PatchCore+ | 75.2 | 97.1 | 78.5 | 97.2 |

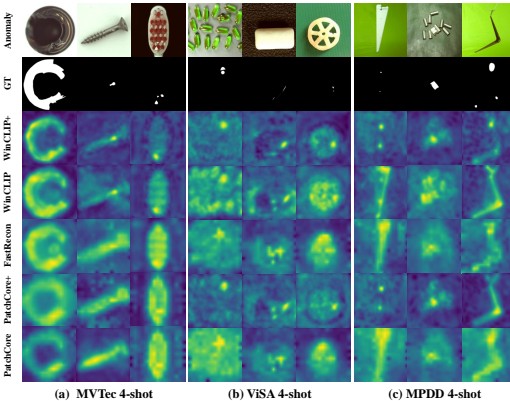

(a) MVTec 4-shot    (b) ViSA 4-shot    (c) MPDD 4-shot

Figure 6: Qualitative results of pixel-level anomaly localization under 4-shot.

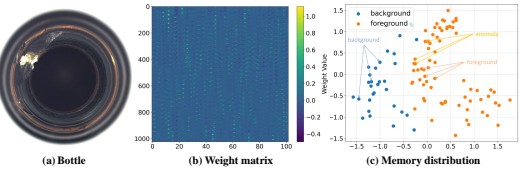

(a) Bottle    (b) Weight matrix    (c) Memory distribution

Figure 7: The learned weight matrix in (b) corresponding to the bottle in (a). (c) shows the Top-3 selected items in the memory according to (b) for different patches of query image in (a).

Table 7: Image/pixel level AUROC and per image inference time (s) on MPDD under 2-shot.

| Method | Image | Pixel | Inference time |
|--------|-------|-------|----------------|
| PatchCore | 71.4 | 96.5 | 0.20 |
| PatchCore+ | 78.2 | 96.9 | 0.50 |
| Closed PatchCore+ | 78.0 | 97.0 | 0.36 |

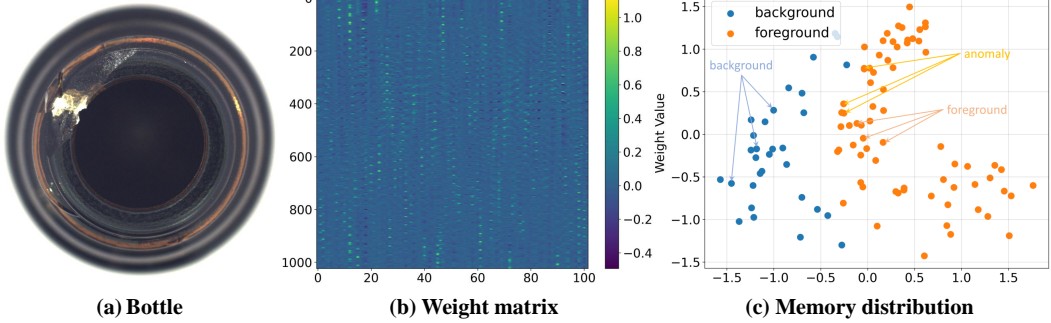

(a) Bottle  (b) Weight matrix  (c) Memory distribution

Figure 8: The learned weight matrix in (b) corresponding to the bottle in (a). (c) shows the Top-3 selected items in the memory according to (b) for different patches of query image in (a).

