# OpenReview forum: "A Prototype-oriented Fast Refinement Model for Few-shot Industrial Anomaly Detection"
_ICLR.cc/2025/Conference — Submitted to ICLR 2025_

### Official Review · Reviewer_Uj7T · 2024-11-01

**Soundness:** 2
**Presentation:** 2
**Contribution:** 1
**Rating:** 5
**Confidence:** 5

**Summary:**

The paper addresses the challenge of industrial anomaly detection (IAD) in low data regimes. The authors argue that previous methods, while constructing normal prototypes from a few normal training images, do not adequately refine these prototypes using query images. Their proposed model leveraging an EM-based optimization algorithm to refine prototypes through iterations that suppress anomalies and transfer characteristics from query features.

**Strengths:**

1.	The paper presents a novel approach to few-shot IAD by formulating prototype refinement as a nested optimization problem, which is innovative in the field of industrial anomaly detection.

2.	The model is designed to integrate existing popular methods like PatchCore and WinCLIP, demonstrating its versatility and potential for widespread adoption.

**Weaknesses:**

1. The motivation of the work is not clear. It lacks necessary explanation on why OT is used, in the EM algorithm.

2. It is necessary to construct experiments to clarify "Point-to-point regularization does significantly limits the ability to transfer characteristics from query images to prototypes" (line 51-52) and “Previous methods may result in suboptimal prototype refinement ……” (line 81-86).

3. Some notations are confusing: $M\in R^{\alpha \times k\times h \times s}$, $s$ is not defined, line 149.

4. The presentation of the article needs to be improved as the current article is difficult to follow.

5. This paper is faithful to the idea of FastRecon, but the advantage of FastRecon is that it directly computes a closed solution for feature reconstruction, which speeds up inference. However, the method proposed by this paper is based on PatchCore, and the actual inference speed is not improved compared with Patchcore and FastRecon, so the term 'Fast' in the title is not justified.

6. Optimal transport is an effective method that can directly achieve significant performance gains, but the authors did not provide further experiments on OT.

**Questions:**

1. In Table 2, is the first setting (w/o W* , T*) naive WinCLIP? Why is the performance lower than the original (Table 1, 2-shot), especially Image-level AUROC?

2. Please provide some explanations on the limited performance improvement of WinCLIP+ over WinCLIP?

---

> ### Author Response · Authors · 2024-11-22
>
> A1: We aim to refine the representativeness of prototypes using statistics of query images while prevent the leakage of anomalous features from query images during inference. Therefore, we propose an optimization framework as shown in Eq. 3 of the manuscript. Specifically, the first term in Eq. 3 is used to refine prototypes in the original memory by transferring statistics of query features. The second term in Eq. 3 is designed to prevent anomaly leakage by minimizing the gap between the refined prototypes and prototypes in the original memory. Considering the fact that we only have the samples from the two sets (the refined prototypes and original prototypes) while their distributions are unknown, a naturally way to evaluate the gap of such kind of two sets shold be OT. Experimental results also verify our motivation and idea.
>
> A2: Comparing the results between FastRecon and ours, it is straightforward to support our claim that "Point-to-point regularization does significantly limits the ability to transfer characteristics from query images to prototypes" (line 51-52) and “Previous methods may result in suboptimal prototype refinement ……” (line 81-86), since FastRecon only stricts point-wise regularization between the mean of refined prototypes and the mean of original prototypes, which might be slightly rough especially in low-data regime.
>
> A3: We change the $\mathcal{\boldsymbol{M}} \in \mathbb{R}^{\alpha \times k \times h \times s}$ to $\mathcal{\boldsymbol{M}} \in \mathbb{R}^{\alpha \times k \times h \times w}$. Additionally, we make more careful proofreading to correct such kind of typos.
>
> A4: We will try our best and consistently improve the quality and readability of our work. And we are glad to hear some specific suggestions such as which parts are difficult for you to follow, and we are glad to make them clear with our common efforts.
>
> A5: (1) We propose a more general optimization framework based on pre-trained features for few-shot IAD task, PatchCore, WinCLIP and even FastRecon could be treated as a special case under our framework. Based on our developed framework, it is naturally to extend PatchCore to PatchCore+ as well as extend WinCLIP to WinCLIP+ as we have discussed in the manuscript. As for FastRecon, its extension version is the same with PatchCore+, thus we do not highlight it in the manuscript. (2) From the optimization perspective, FastRecon only stricts point-wise regularization between the mean of refined prototypes and the mean of original prototypes, which might be slightly rough especially in low-data regime. On the contrary, our model prevent the anomaly leakage from query features to the refined prototypes in a distribution matching manner. In order to achieve this goal, we also develop an efficient EM-based iterative optimization algorithm. (3) Fast should be a tradeoff between precise and running time. Additionally, as suggested by reviewer MiAm, we further derive a closed form solution for transform matrix $\boldsymbol{W}$, which makes our optimization even faster in updating transform matrix $\boldsymbol{W}$ and transport matrix $\boldsymbol{T}$ iteratively. Results are reported in Table 7 of the appendix. For convience, we also show Table 7 of the appendix as follows:
>
> | Method              | Image AUROC | Pixel AUROC | Inference Time (s) |
> |:--------------------|:-----------:|:-----------:|:------------------:|
> | PatchCore           | 71.4        | 96.5        | 0.20              |
> | PatchCore+          | 78.2        | 96.9        | 0.50              |
> | Closed PatchCore+   | 78.0        | 97.0        | 0.36              |
>
> A6: The ablation studies on transform matrix and transport matrix (OT) are illustrated in Table 2 of the manuscript.
>
> Q1: We have checked our implementation and find that the sets of support images used to report the results in the first line of Table 2 are different from those in the second and the third lines of Table 2. We have modified the mistake and report the updated results in Table 5 of the appendix. And we will release our code for reproduction upon acceptance. For convinence, we report Table 5 of the appendix as follows:
>
> | $W^*$       | $T^*$       | MVTec Image | MVTec Pixel | VisA Image | VisA Pixel | MPDD Image | MPDD Pixel |
> |:-------------:|:-------------:|:-----------:|:-----------:|:----------:|:----------:|:----------:|:----------:|
> | **$\times$**  | **$\times$**  | 93.7        | 93.8        | 83.8       | 95.1       | 72.5       | 96.5       |
> | **$\checkmark$** | **$\times$** | 93.7        | 94.7        | 83.7       | 96.2       | 74.9       | 96.4       |
> | **$\checkmark$** | **$\checkmark$** | **93.9**   | **96.2**   | **84.1**  | **96.4**  | **76.0**  | **97.3**  |

---

> ### Author Response · Authors · 2024-11-22
>
> Q2: Comparing results of WinCLIP and PatchCore in few-shot IAD under MVTec and ViSA, we guess that the differences mainly come from backbone pre-training. As we know that backbone pre-training of WinCLIP is much better than that of PatchCore. Therefore, our prototye refinement provides more benefit when integrated with PatchCore than WinCLIP. On the other hand, according to the results on MPDD in Table 1 of the manuscript, we notice that the gains from WinCLIP to WinCLIP+ increase. And we attribute this to that the backbone of WinCLIP has rarely been pre-trained on such metal objects with rotation, echoing the previous conjecture again.

---

> > ### Author Response · Authors · 2024-11-23
> >
> > Dear Reviewer Uj7T,
> >
> > We appreciate it if you could let us know whether our responses are able to address your concerns. We're happy to address any further concerns. Thank you,
> >
> > Best wishes.
> >
> > Paper 5785 Authors

---

> > > ### Comment · Reviewer_Uj7T · 2024-11-25
> > >
> > > The authors have largely addressed my concerns. However, the approach combining memory bank and optimal transport (OT) for few-shot anomaly detection was previously proposed in reference [1] (There are numerous striking similarities). Additionally, the results in Table 7 demonstrate that this method does not provide significant improvements in inference speed. Given these considerations, the contributions of this work are limited. Therefore, I maintain the original rating score while increasing the confidence level.
> > >
> > > [1] Long Tian, Hongyi Zhao, Ruiying Lu, Rongrong Wang, Yujie Wu, Liming Wang, Xiongpeng He, and Xiyang Liu. 2024. FOCT: Few-shot Industrial Anomaly Detection with Foreground-aware Online Conditional Transport. In Proceedings of the 32nd ACM International Conference on Multimedia (MM '24).

---

> ### Author Response · Authors · 2024-11-25
>
> Dear reviewer Uj7T,
>
> Thanks for your reply.
>
> (1) We claim that our model is a novel optimization framework which is efficient for solving few-shot IAD inspired by the idea of test-time training. Our model differs from [1] mainly in three aspects as follows:
>
> i) the optimization objective is totally different.
>
> In this work, our objective is
>
> $\mathcal{L}(\boldsymbol{W}, \boldsymbol{T})=dis(\boldsymbol{f}_t^q, \boldsymbol{W} \mathcal{\boldsymbol{M}}_s)+\lambda OT(\boldsymbol{W} \mathcal{\boldsymbol{M}}_s, \mathcal{\boldsymbol{M}}_s)$
>
> Namely, we want to learn a transform matrix to refine prototypes by composition according to the normal memory via aligning distributions between refined memory and normal memory. By using the refined prototypes (composition generalization) we implement IAD under Euclidean or Cosine distance measurement. However, In [1], their objective can be expressed as
>
> $\mathcal{L}_{\boldsymbol{\phi}, \rho}=CT(\boldsymbol{f}_t^q, \mathcal{\boldsymbol{M}}_s)$
>
> where $CT(.,.)$ is conditional transport [1] that parameterizing the transport matrix $\boldsymbol{T}$ used in OT with neural networks ($\boldsymbol{\phi}$), MLP for example. It is obvious that in [1], the authors aim to select query features that are most relevant with the normal memory by aligning distributions between query features and normal memory. By using the augmented memory (augment with selected query features) they conduct IAD under CT distance measurement.
>
> ii) The objective properties are different.
>
> The objective of our model with entropy relaxation is weak convex, demonstration could be found in Sinkhorn algorithm [2]. However, the objective of [1] is non-convex since MLP's nonlinear property.
>
> iii) The optimization methods are totally different.
>
> In [1], the neural networks  ($\boldsymbol{\phi}$)  can only be optimized via SGD to find local optimization. On the contrary, thanks to the weak convex property of our objective, in our work, parameters such as transport matrix ($\boldsymbol{T}$) and transform matrix ($\boldsymbol{W}$) could be solved efficiently in closed form to find global optimization.
>
> [1] FOCT: Few-shot Industrial Anomaly Detection with Foreground-aware Online Conditional Transport.
>
> [2] Sinkhorn Distances: Lightspeed Computation of Optimal Transport.
>
> (2) As for inference speed, test-time training itself is a balanced between performance and inference time, which is also employed in nowadays AIGC finetuning. Therefore, we argue that an algorithm can not only be judged by its inference speed. Meanwhile, our model achieves improved performance while only cost a little more inference time. Hence, we hope that the results could be reconsidered by reviewer Uj7T. Especially when we update $\boldsymbol{W}$ in a closed form as reviewer MiAm suggested, the inference speed is further significantly improved, as shown in Table 7 of the appendix. Additionally, if we want to deploy the model on the actual production line, other acceleration technologies could also be adopted. To sum up, we do not wish that the decision of reviewer Uj7T be restricted by inference speed of our model (since we have considered it and made optimization on it effectively). For convienence, we report Table 7 of the appendix as follows:
>
> **Table 7:** Image/Pixel level AUROC and per image inference time on MPDD under 2-shot. Closed PatchCore+ denotes an accelerated version by solving $\boldsymbol{W}$ with a closed form.
>
> | Method              | Image AUROC | Pixel AUROC | Inference Time (s) |
> |:--------------------|:-----------:|:-----------:|:------------------:|
> | PatchCore           | 71.4        | 96.5        | 0.20              |
> | PatchCore+          | 78.2        | 96.9        | 0.50              |
> | Closed PatchCore+   | 78.0        | 97.0        | 0.36              |

---

> > ### Author Response · Authors · 2024-11-25
> >
> > Dear Reviewer Uj7T,
> >
> > Given that the rebuttal phase is about to end, we would like to know if our responses have addressed your concerns. Additionally, we are still here and happy to address any further concerns.
> >
> > Best wishes,
> >
> > Paper 5785 Authors

---

> > ### Author Response · Authors · 2024-11-27
> >
> > Dear Reviewer Uj7T,
> >
> > We appreciate it if you could let us know whether our responses are able to address your concerns. We're happy to address any further concerns. Thank you,
> >
> > Best wishes.
> >
> > Paper 5785 Authors

---

> > > ### Comment · Reviewer_Uj7T · 2024-11-28
> > >
> > > From a high-level perspective, this work shares similar objectives with FOCT (Few-shot Industrial Anomaly Detection with Foreground-aware Online Conditional Transport) in obtaining more suitable prototypes, and both methods utilize Optimal Transport (OT) for memory bank refinement. The use of Sinkhorn algorithm to solve OT represents only an incremental improvement. Given that the authors propose a "Fast Refinement Model" as indicated in the title, inference time should be carefully considered. While the authors' discussion of the trade-off between inference time and accuracy is reasonable, the proposed method does not
> > >  demonstrate superior inference time while maintaining competitive (not SOTA) accuracy. Therefore, I finally decided to keep the original grade.

---

> ### Author Response · Authors · 2024-11-28
>
> Thanks for your reply,
>
> (1) **From a high-level perspective**, in FOCT, they utilize Conditional Transport (CT), instead of Optimal Transport (OT) as reviewer understood, to select features most likely to be normal from query features. However, in our work, we propose an OT-based objective to realize composition generalization from normal support features, which is significantly different from FOCT. Besides, CT is based on MLP to learn transport matrix, which is computational complex compared with our closed form. Therefore, we still persevere that our method is significantly different from FOCT. Additionally, we do not agree with the reviewer that our method do not demonstrate superior inference time while not SOTA. At test time, if you want to obtain performance gains by leveraging statistics of query features, it is usually inevitable to spend more time. And we do not feel this should be blamed. And we are willing to modify our title if "Fast" really bothers you.
> As for SOTA, our method achieves SOTA or competitive results on MVTec, ViSA, MPDD, and RealIAD according to our experimental results. What is important is that our model is the first work to unify memory-based few-shot IAD with an optimization framework as far as we know, which also achieves consistently performance improvements compared with their original methods such as PatchCore, FastRecon, and WinCLIP.
>
> (2) There are significant differences in CT and OT, not only in objective but also in their properties. Please refer to [1] and [2] for details.
>
> [1] Exploiting Chain Rule and Bayes' Theorem to Compare Probability Distributions.
>
> [2] Sinkhorn Distances: Lightspeed Computation of Optimal Transport.
>
> We appreciate it if you could let us know whether our responses are able to address your concerns. We're happy to address any further concerns. Thank you,
>
> Best wishes.
>
> Paper 5785 Authors

---

> > ### Author Response · Authors · 2024-12-01
> >
> > Dear Reviewer Uj7T,
> >
> > Given that the rebuttal phase is about to end, we would like to know if our responses have addressed your concerns. Additionally, we are still here and happy to address any further concerns.
> >
> > Best wishes,
> >
> > Paper 5785 Authors

---

> ### Author Response · Authors · 2024-12-01
>
> Dear reviewer Uj7T,
>
> We have requested and obtained the official codes from the authors of FOCT and implemented comparisons of our model and FOCT on MVTec, ViSA, and MPDD under 2 shot, results are reported as follows:
>
> | Methods       | MVTec Image | MVTec Pixel | VisA Image | VisA Pixel | MPDD Image | MPDD Pixel |
> |:-------------:|:-----------:|:-----------:|:----------:|:----------:|:----------:|:----------:|
> | FOCT  | 90.5        | 94.8        | 86.3       | 95.9       | **82.4**       | 96.5       |
> | PatchCore+ (ours) | 88.8        | 94.7        | **87.1**       | **98.0**       | 78.2       | 96.9       |
> | WinCLIP+ (ours) | **93.9**        | **96.2**        | 84.1       | 96.4       | 76.0       | **97.3**       |
>
> As we can see that our models (PatchCore+ and WinCLIP+) achieve better results (5/6, marked in bold) compared with FOCT. Meanwhile, FOCT leverages both statistics of current query features and historical query features while our model only use statistics of current query features. It verifies superority of our model for composition generalization from support  features in few-shot IAD task.
>
> We will also replenish comparison results with FOCT in the final version of manuscript.
>
> We appreciate it if you could let us know whether our responses are able to address your concerns. We're happy to address any further concerns. Thank you,
>
> Best wishes.
>
> Paper 5785 Authors

---

> ### Author Response · Authors · 2024-12-02
>
> Dear Reviewer Uj7T,
>
> We sincerely appreciate your insightful comments and the acknowledgment of our paper from the real-time efficiency and presentation sides. As the rebuttal deadline approaches, we would like to know whether there are any additional concerns we can address. We will try our best to provide further clarification.
>
> Few-shot IAD is a promising but challenging setting where most current works try to avoid it resulting in limited generalizability. The proposed prototype refinement method is the first work managing to provide a unified view for solving few-shot IAD efficiently and effectively, which is also a flexible method for solving other prototype-oriented tasks like video anomaly detection and prediction (Please refer to response for reviewer Exqa). None of the above-mentioned will be possible without your support.
>
> We hope our provided response during the rebuttal addresses the reviewers' concerns and the reviewers can re-evaluate our paper. We thank the reviewers once again for their efforts.
>
> Best regards,
> Authors of Paper 5785

---

### Official Review · Reviewer_HDry · 2024-11-01

**Soundness:** 3
**Presentation:** 3
**Contribution:** 3
**Rating:** 6
**Confidence:** 5

**Summary:**

This paper proposed an enhancenment module for feature similarity measurement based Anomaly Detection method and conducted experiments on Few-shot Anomaly Detection (AD) setting, a topic that has recently gained considerable attention and holds practical value in real-world scenarios. The authors' innovative use of optimal transport to optimize the process of feature distance measurement is commendable, as it not only introduces novelty but also demonstrates generality by being applicable to different methods. Additionally, the overall organization and writing of the article are well-executed, making it easy to follow and understand. However, the paper has some weaknesses. There is a notable lack of experimental comparisons; for instance, if the GraphCore method published in ICLR23 achieves significantly higher P-AUROC for MVTec 4-shot and I-AUROC for MPDD 4-shot compared to the best results of the paper's method, these results should be included in the comparison table (Table 1). The analysis of experimental results also needs to be strengthened, as there is significant variation in the improvement observed across different datasets. The proposed method shows a larger improvement on WinClip in MVTec but a larger improvement on PatchCore in VisA, and this inconsistency lacks theoretical analysis. The methods described in the article can be seen as a plugin for enhancing performance, and directly showing the increment in Table 1 would make it easier to read. Especially when the increment varies across different datasets, calculating the average improvement could better reflect the value of the method. Additionally, there is an inconsistency in the font used for "l" in Formula (1) on line 144, as well as lines 145 and 146, which can lead to confusion. Lastly, the relative changes in results on MPDD are quite noticeable, which could be attributed to the smaller size of the MPDD dataset (458 test images) and its inherent randomness. Conducting comparative experiments on larger datasets, such as the newly proposed Real-IAD dataset in CVPR2024, would provide a better reflection of the effectiveness of the methods.

**Strengths:**

1.Few-shot AD, which has gained considerable attention recently, also holds practical value in real-world scenarios.
2.Leveraging optimal transport to optimize the process of feature distance measurement is innovative and also exhibits generality in being grafted onto different methods.
3.The overall organization and writing of the article are well done.

**Weaknesses:**

1.There seems to be a lack of experimental comparisons. If the GraphCore method published in ICLR23 achieves significantly higher P-AUROC for MVTec 4-shot and I-AUROC for MPDD 4-shot compared to the best results of the paper's method, it should have been included in the comparison table (Table 1).

2.The analysis of experimental results needs to be strengthened. There is a significant variation in the improvement observed across different datasets. The proposed method shows a larger improvement on WinClip in MVTec, but a larger improvement on PatchCore in VisA. This inconsistency lacks theoretical analysis.

3. The methods described in the article can be seen as a plugin for enhancing performance. In the comparison with other methods in Table 1, directly showing the increment in the table makes it easier to read. Especially when the increment varies across different datasets, it may be better to calculate the average improvement to reflect the value of the method.

4.  inconsistency in the font used for "l" in Formula (1) on line 144, as well as lines 145 and 146. It can indeed lead to confusion.

Review information added after rebutal:
The weakness of 1, 4 has been explained, and authors will fix them in the final version after acceptance.

**Questions:**

1. The relative changes in results on MPDD are quite noticeable, which could be attributed to the smaller size of the MPDD dataset (458 test images) and its inherent randomness. Conducting comparative experiments on larger datasets, such as the newly proposed Real-IAD dataset in CVPR2024, would provide a better reflection of the effectiveness of the methods.

Review information added after rebutal:
My concerns have been addressed, and the author will add the experiment results in the final version after acceptance.

---

> ### Author Response · Authors · 2024-11-22
>
> A1: We add performance comparisons of our model and GraphCore in Table 4 of the appendix. Although the performance of GraphCore is competitive, our methods exceeds GraphCore in most cases. And we will also update the Table 1 in the final version of manusript. For conveniene, we report Table 4 of the appendix as follows:
>
> | Setup   | Method                 | MVTec Image | MVTec Pixel | VisA Image | VisA Pixel | MPDD Image | MPDD Pixel | RealIAD Image | RealIAD Pixel |
> |---------|------------------------|-------------|-------------|------------|------------|------------|------------|---------------|---------------|
> | 1-shot  | GraphCore (ICLR'23)    | 89.9        | **95.6**    | ---        | ---        | **84.7**   | 95.2       | ---           | ---           |
> |         | PatchCore (CVPR'22)    | 84.1        | 92.3        | 71.0       | **96.1**   | 71.0       | 96.3       | 71.2           | 95.7           |
> |         | PatchCore+ (Ours)      | 85.9 (+1.8) | 93.7 (+1.4) | 78.3 (+7.3)| 97.1 (+1)  | **74.9 (+3.9)** | **96.6 (+0.3)** | **75.9(+4.7)**       | **96.3(+0.6)**           |
> |         | WinCLIP (CVPR'23)      | **93.5**    | 93.6        | **83.4**   | 94.7       | 70.5       | 96.3       | 73.8           | 94.3           |
> |         | WinCLIP+ (Ours)        | **93.8 (+0.3)** | **95.7 (+2.1)** | **83.9 (+0.5)** | 95.8 (+1.1) | 72.5 (+2.0) | **96.9 (+0.6)** | 74.4(+0.6)       | 94.8(+0.5)           |
> |---------|------------------------|-------------|-------------|------------|------------|------------|------------|---------------|---------------|
> | 2-shot  | GraphCore (ICLR'23)    | 91.9        | **96.9**    | ---        | ---        | **85.4**   | 95.4       | ---           | ---           |
> |         | PatchCore (CVPR'22)    | 87.1        | 93.3        | 80.0       | **96.9**   | 71.4       | 96.5       | 72.5          | **95.9**      |
> |         | PatchCore+ (Ours)      | 88.8 (+1.7) | 94.7 (+1.4) | **87.1 (+7.1)** | **98.0 (+1.1)** | **78.2 (+6.8)** | **96.9 (+0.4)** | **76.9 (+4.4)** | **96.5 (+0.6)** |
> |         | WinCLIP (CVPR'23)      | **93.7**    | 93.8        | 83.8       | 95.1       | 72.5       | 96.5       | 75.0          | 94.6          |
> |         | WinCLIP+ (Ours)        | **93.9 (+0.2)** | **96.2 (+2.4)** | **84.1 (+0.3)** | 96.4 (+1.3) | 76.0 (+3.5) | **97.3 (+0.8)** | **75.9 (+0.9)** | 95.2 (+0.6) |
> |---------|------------------------|-------------|-------------|------------|------------|------------|------------|---------------|---------------|
> | 4-shot  | GraphCore (ICLR'23)    | 92.9        | **97.4**    | ---        | ---        | **85.7**   | 95.7       | ---           | ---           |
> |         | PatchCore (CVPR'22)    | 90.0        | 95.1        | 84.2       | **97.5**   | 76.2       | 97.2       | 73.2           | 96.0           |
> |         | PatchCore+ (Ours)      | 92.1 (+2.1) | 96.1 (+1.0) | **90.4 (+6.2)** | **98.2 (+0.7)** | 80.3 (+4.1) | **97.2 (+0)**  | **77.4(+4.2)**          | **96.7(+0.7)**           |
> |         | WinCLIP (CVPR'23)      | **95.3**    | 94.2        | 84.1       | 95.4       | 75.0       | 96.8       | 76.4           | 94.8           |
> |         | WinCLIP+ (Ours)        | **95.5 (+0.2)** | **96.7 (+2.5)** | **85.0 (+0.9)** | 96.6 (+1.2) | **82.0 (+7)** | **97.6 (+0.8)** | 77.3(+0.9)       | 95.3(+0.5)           |
>
>
> A2: Our model is a plug-and-play module based on some well pre-trained backbones, such as WRN and CLIP separately used in PatchCore and WinCLIP. To better understand the effectiveness and efficiency of our proposed two methods, PatchCore+ and WinCLIP+, the improvements compared with their corresponding base models, PatchCore and WinCLIP, should be crucial. As for variation across different datasets, we guess it might be attributed to the pre-training stage and we plan to leave it in the future since it’s beyond our current work.
>
> A3: We replenish the performance with corresponding increment compared with its base models and update the results in Table 4 of the appendix. Also refer to Table 4 reported in A1.
>
> A4: We correct the font inconsistency of the two “l” from line 144 to 146 and make more careful proofreading for the details.
>
> A5: We verify the effectiveness our model on Real-IAD and replenish results under 2-shot in Table 4 of the appendix. Also see Table 4 reported in A1. Generally, our model achieves significant improvements compared with its baselines. We will also provide more detailed comparisons and analysis under 1/4-shot in the final version of manuscript.

---

> ### Comment · Reviewer_HDry · 2024-11-22
>
> General, the authors addressed some of my concerns. But there is some confusing points.
>
> 1. Experiments has been extended to newly proposed Real-IAD dataset, and under 2-shot setting, there is an significant improvement on PatchCore, and has some improvement on WinClip. But I am wondering why there is only results on 2-shot, while results on 1-shot and 4-shot are not reported.
>
> 2. About the comparison with GraphCore(ICLR'23).  The proposed method has its advantages and disadvantage compared with GraphCore under different experiment settings. It would be better to make an average performance to clearly show which is better.

---

> > ### Author Response · Authors · 2024-11-22
> >
> > Thanks for your reply.
> >
> > 1. We are still conducting experiments on 1-shot and 4-shot since we only have a single NVIDIA GTX 3090 GPU and Real-IAD is really a big dataset in this situation, the updated results will coming soon. Thanks for your patience.
> >
> > 2. We average the performance of our method and GraphCore along different datasets. For example, we average the Image level AUROC of our best model on MVTec under 1/2/4-shot, and the same for GraphCore. We report results as follows:
> >
> > | Method              | MVTec Image | MVTec Pixel | MPDD Image | MPDD Pixel |
> > |:--------------------|:------------:|:------------:|:------------:|:------------:|
> > | GraphCore           | 91.6         | 96.6         | 85.3         | 95.4         |
> > | Ours(best)          | 94.4         | 96.2         | 78.4         | 97.3         |
> >
> > As we can see that our method could achieve competitive results in general. It is worth to say that GraphCore is trained on these datasets, which we believe could bring it more benefit especially for metal targets with rotation like MPDD. Additionally, GraphCore is also a memeory-based few-shot IAD model, namely, it should also be further enhanced by our proposed framework just like PatchCore and WinCLIP do. Unfortunately, the official code right now is still unavaliable. And we have to leave it as our future work.

---

> > > ### Comment · Reviewer_HDry · 2024-11-25
> > >
> > > The auhtors have addressed most of my concerns. The authors demonstrated the effectiveness of the method on the newly published more challenging dataset Real-IAD. And showed the advantages over GraphCore. I will raise the rating of the paper.

---

> > ### Author Response · Authors · 2024-11-24
> >
> > Dear reviewer HDry,
> >
> > As you suggested, we have implemented experiments on Real-IAD dataset under 1/2/4-shot settings and reported the results in Table 4 of the appendix. Our method achieves consistent improvement on Real-IAD. For convienence, we also report Table 4 as follows:
> >
> > | Setup   | Method                 | MVTec Image | MVTec Pixel | VisA Image | VisA Pixel | MPDD Image | MPDD Pixel | RealIAD Image | RealIAD Pixel |
> > |---------|------------------------|-------------|-------------|------------|------------|------------|------------|---------------|---------------|
> > | 1-shot  | GraphCore (ICLR'23)    | 89.9        | **95.6**    | ---        | ---        | **84.7**   | 95.2       | ---           | ---           |
> > |         | PatchCore (CVPR'22)    | 84.1        | 92.3        | 71.0       | **96.1**   | 71.0       | 96.3       | 71.2           | 95.7           |
> > |         | PatchCore+ (Ours)      | 85.9 (+1.8) | 93.7 (+1.4) | 78.3 (+7.3)| 97.1 (+1)  | **74.9 (+3.9)** | **96.6 (+0.3)** | **75.9(+4.7)**       | **96.3(+0.6)**           |
> > |         | WinCLIP (CVPR'23)      | **93.5**    | 93.6        | **83.4**   | 94.7       | 70.5       | 96.3       | 73.8           | 94.3           |
> > |         | WinCLIP+ (Ours)        | **93.8 (+0.3)** | **95.7 (+2.1)** | **83.9 (+0.5)** | 95.8 (+1.1) | 72.5 (+2.0) | **96.9 (+0.6)** | 74.4(+0.6)       | 94.8(+0.5)           |
> > |---------|------------------------|-------------|-------------|------------|------------|------------|------------|---------------|---------------|
> > | 2-shot  | GraphCore (ICLR'23)    | 91.9        | **96.9**    | ---        | ---        | **85.4**   | 95.4       | ---           | ---           |
> > |         | PatchCore (CVPR'22)    | 87.1        | 93.3        | 80.0       | **96.9**   | 71.4       | 96.5       | 72.5          | **95.9**      |
> > |         | PatchCore+ (Ours)      | 88.8 (+1.7) | 94.7 (+1.4) | **87.1 (+7.1)** | **98.0 (+1.1)** | **78.2 (+6.8)** | **96.9 (+0.4)** | **76.9 (+4.4)** | **96.5 (+0.6)** |
> > |         | WinCLIP (CVPR'23)      | **93.7**    | 93.8        | 83.8       | 95.1       | 72.5       | 96.5       | 75.0          | 94.6          |
> > |         | WinCLIP+ (Ours)        | **93.9 (+0.2)** | **96.2 (+2.4)** | **84.1 (+0.3)** | 96.4 (+1.3) | 76.0 (+3.5) | **97.3 (+0.8)** | **75.9 (+0.9)** | 95.2 (+0.6) |
> > |---------|------------------------|-------------|-------------|------------|------------|------------|------------|---------------|---------------|
> > | 4-shot  | GraphCore (ICLR'23)    | 92.9        | **97.4**    | ---        | ---        | **85.7**   | 95.7       | ---           | ---           |
> > |         | PatchCore (CVPR'22)    | 90.0        | 95.1        | 84.2       | **97.5**   | 76.2       | 97.2       | 73.2           | 96.0           |
> > |         | PatchCore+ (Ours)      | 92.1 (+2.1) | 96.1 (+1.0) | **90.4 (+6.2)** | **98.2 (+0.7)** | 80.3 (+4.1) | **97.2 (+0)**  | **77.4(+4.2)**          | **96.7(+0.7)**           |
> > |         | WinCLIP (CVPR'23)      | **95.3**    | 94.2        | 84.1       | 95.4       | 75.0       | 96.8       | 76.4           | 94.8           |
> > |         | WinCLIP+ (Ours)        | **95.5 (+0.2)** | **96.7 (+2.5)** | **85.0 (+0.9)** | 96.6 (+1.2) | **82.0 (+7)** | **97.6 (+0.8)** | 77.3(+0.9)       | 95.3(+0.5)           |

---

> > > ### Author Response · Authors · 2024-11-24
> > >
> > > Dear Reviewer HDry,
> > >
> > > We appreciate it if you could let us know whether our responses are able to address your concerns. We're happy to address any further concerns. Thank you,
> > >
> > > Best wishes.
> > >
> > > Paper 5785 Authors

---

> > > > ### Comment · Reviewer_HDry · 2024-11-26
> > > >
> > > > A small tip:
> > > >
> > > > Experiment description and Reference should be revised according to the modifications during rebutal phase if the paper is finally accepted.

---

> ### Author Response · Authors · 2024-11-27
>
> Dear reviewer HDry,
>
> Thanks for your reply and valuable suggestions, we will revise experiment description and reference according to the modifications in our final version of manuscript and try our best to improve the quality of the paper.  And We're happy to address any further concerns. Thank you,
>
> Best wishes.
>
> Paper 5785 Authors

---

### Official Review · Reviewer_Exqa · 2024-11-02

**Soundness:** 2
**Presentation:** 3
**Contribution:** 2
**Rating:** 6
**Confidence:** 4

**Summary:**

To address the issue of prototypes not adequately representing the features of few-shot anomaly detection, the authors propose a prototype-oriented fast refinement model. They transform the problem of prototype refinement into a nested optimization problem and introduce an Expectation Maximization (EM)-based algorithm to iteratively compute the transport probability and transform matrix. The Sinkhorn algorithm is used to learn the transport probability. The effectiveness of the proposed method is demonstrated on PatchCore and WinCLIP, as well as on three anomaly detection datasets: MVTec, ViSA, and MPDD, under different shot settings.

**Strengths:**

1. The authors propose transforming the problem of prototype refinement into a nested optimization problem and solving it through optimal transport, which is a novel approach.

2. The authors achieve significant performance improvements on PatchCore and WinCLIP across different shot settings on the MVTec, ViSA, and MPDD anomaly detection datasets.

3. The authors demonstrate the efficiency of the method.

**Weaknesses:**

1. In Figure 3, it is recommended to simultaneously display the qualitative experimental results of PatchCore+ to enhance reliability.

2. It is suggested that the authors provide code or reliable PyTorch pseudocode.

3. Besides being extendable to PatchCore and WinCLIP, is the proposed structure applicable to other methods? The authors should provide explanations or experiments to demonstrate its extensibility and transferability.

4. The paper has some structural issues, notably the absence of a conclusion section.

**Questions:**

See Weaknesses.

---

> ### Author Response · Authors · 2024-11-22
>
> A1: We replenish the visualization of PatchCore+ in Figure 6 of the appendix. And we will also add it in the final manuscript.
>
> A2: We will release the code upon acceptance. And the pseudocode is illustrated in Algorithm 1 of the appendix. For convenience, we also illustrate algorithm 1 of the appendix as follows:
>
> ---
> ### **Algorithm 1**: Inference Process
> ---
>
> **Require**:  Initial transform matrix $W_0$, original prototypes $\boldsymbol{\mathcal{M}}_{\rm{s}}$, the $t$-th query features $f_t^q$
>
> 1. **Calculating** $p_s$ using $\boldsymbol{\mathcal{M}}_{\rm{s}}$
>
> 2. **for** $m = 0$ to $M-1$:
>    - **Calculating** $q_s$ using $W_m \boldsymbol{\mathcal{M}}_{\rm{s}}$
>    - **for** $e = 0$ to $E-1$:
>      - **Update** $T_{m+1}$ from $T_m$ by minimizing:
>        $$\text{OT}(p_s, q_s) \text {  while fix  } W_m $$
>    - **Update** $W_{m+1}$ from $W_m$ by minimizing:
>
>       $\mathcal{L}$ ( $f_t^q, \boldsymbol{\mathcal{M}}\_{\rm{s}};$ $W,$ $T$) $\text{  while fix  }$ $T_{m+1} $
>
> 3. **Set** $W^* = W_M,$ $\boldsymbol{\mathcal{M}}\_{\rm{s}}^\* = $  $ W\^\*$$\boldsymbol{\mathcal{M}}_{\rm{s}}$
>
> 4. **Implement** few-shot IAD as follows:
>    $ \boldsymbol{\mathcal{s}}\_{\rm{j}} := $ $\min_{\boldsymbol{r} \in \boldsymbol{\mathcal{M}}\_{\rm{s}}^\*} \text{dis}(f_{\rm{t},\rm{j}}^{\rm{q}}, \boldsymbol{r}),  \text{     }{\rm{j}} = 1, \dots, {\rm{m}} $
> ---
>
> A3: Theoretically speaking, our model could be extended to any memory-based few-shot IAD methods, including PatchCore and WinCLIP. Additionally, our model can also be used to extend FastRecon [1]. However, the extended version of FastRecon is same with PatchCore+, hence, we do not highlight it in the manuscript.
> And we will also highlight the extension of our model based on FastRecon in the final manuscript.
> As for other memory-based IAD methods, they usually deal with IAD with abundant training images, which is no longer few-shot scenario.
>
> [1] FastRecon: Few-shot Industrial Anomaly Detection via Fast Feature Reconstruction.
>
> A4: We will replenish conclusion in the final version of manuscript. The conclusion is as follows:
>
>  This paper addresses the problem of industrial anomaly detection (IAD) in low-data regions by proposing a prototype-oriented few-shot IAD rapid refinement model. Given an online query image, the model formulates prototype refinement as a nested optimization problem between the transport probability for anomaly suppression and the transformation matrix for feature transfer. An expectation-maximization (EM)-based algorithm iteratively computes the transport probability and transformation matrix to optimize the prototype and enhance anomaly detection performance. Integrated experiments with two popular few-shot IAD methods, PatchCore and WinCLIP, validate the effectiveness and efficiency of the proposed model in few-shot IAD applications on three widely used datasets, including MVTec, ViSA, and MPDD. This work provides a new approach for prototype-based few-shot anomaly detection in low-data regions.

---

> > ### Author Response · Authors · 2024-11-23
> >
> > Dear Reviewer Exqa,
> >
> > We appreciate it if you could let us know whether our responses are able to address your concerns. We're happy to address any further concerns. Thank you,
> >
> > Best wishes.
> >
> > Paper 5785 Authors

---

> > > ### Comment · Reviewer_Exqa · 2024-11-24
> > >
> > > The rebuttal phase provides ample time to conduct few-shot experiments that do not require training. Therefore, I am still concerned about the results and decide to keep the original score.

---

> > > > ### Author Response · Authors · 2024-11-24
> > > >
> > > > Dear reviewer Exqa,
> > > >
> > > > Thanks for your reply. (1) As we discussed before that memory-based few-shot IAD methods are limited, the most popular ones with official codes are PatchCore, WinCLIP, and FastRecon as far as we know. Therefore, we have conducted experiments on them in the manuscript. (2) If you have any suggested memory-based few-shot IAD methods that we missed, please let we know and we are glad to replenish results on them. Besides, we will also try to find some other new and avaliable memory-based few-shot IAD methods to echo your suggestions in Weakness 3.

---

> > > > ### Author Response · Authors · 2024-11-27
> > > >
> > > > Dear reviewer Exqa,
> > > >
> > > > Thanks for your patience again. We have spent several days to search associated extension of our model in a broader range of memory-based anomaly detection task and decided to further verify the extensibility and transferability of our model on memory-based video anomaly detection. We employ a competitive baseline method called MNAD [1] published on CVPR 2020 and integrate our model as a plug-ins to enhance the anomaly detection capability of MNAD at test time. Specifically, we first train MNAD for 100 epochs with default config of the authors using UCSD Ped2 dataset [2]. Then, we replaced the read mechanism of MNAD in Eq. 1 and 2 of the paper with our model during inference. The performance is evaluated under the task of anomaly prediction, that is, we use four successive video frames to predict the fifth one. Results are reported as follows:
> > > >
> > > > **Table:** AUROC (\%) on UCSD Ped. The best result is marked in bold.
> > > >
> > > > | Method              | Ped2 |
> > > > |:--------------------|:------------:|
> > > > | Frame-Pred           | 95.4         |
> > > > | MNAD          | 96.2         |
> > > > | MNAD+ours   | **96.8**     |
> > > >
> > > > [1] Learning Memory-guided Normality for Anomaly Detection.
> > > > [2] Anomaly Detection and Localization in Crowded Scenes.
> > > >
> > > > We hope that our response with additional experiments could address your concerns. And we're happy to address any further concerns. Thank you,
> > > >
> > > > Best wishes.
> > > >
> > > > Paper 5785 Authors

---

### Official Review · Reviewer_MiAm · 2024-11-04

**Soundness:** 3
**Presentation:** 2
**Contribution:** 2
**Rating:** 5
**Confidence:** 4

**Summary:**

The paper proposes a prototype-oriented fast refinement model for few-shot industrial anomaly detection. It focuses on enhancing anomaly detection under few-shot condition by improving prototype representativeness with query image features. It formulates prototype refinement as a nested optimization problem, balancing anomaly suppression and characteristic transfer. The problem is solved iteratively with an EM-based algorithm. The model integrates with few-shot anomaly detection methods like PatchCore and WinCLIP, showing significant improvements in anomaly detection performance on datasets such as MVTec, ViSA, and MPDD

**Strengths:**

- Few-shot anomaly detection could have high impact in industrial anomay detection tasks. The proposed method improved the efficacy of few-shot industrial anomaly detection by combining with state-of-the-art methods.

- Extensive evaluations are carried out on three industrial anomaly detection datasets.

**Weaknesses:**

- The optimization task in Eq. (3) is not well explained. The first term minimizes the distance between transformed prototypes, $WM_s$ and query features. While the optimal transport also measures the distance between prototypes and the query sample, subject to an optimal assignment. The two terms seem to overlap. A more thorough discussion on the difference and why the optimal transport regularization is necesssary is required.

- The prototype transformation acts more like a prototype selector. As shown in Eq (7), $W$ is first optimized to minimize the difference between query feature and selected prototypes. Moreover, the prototypes do not benefit from observing more testing/query samples. Therefore, the proposed step does not refine the prototype, but rather selects the most appropriate prototypes for the given query feature/sample.

- In the EM algorithm, I am wondering why gradient descent is adopted for update $W$. Since OT is independent of $W$, the optimization w.r.t. $W$ in Eq (7) is a simple linear regression problem. A closed-form solution exists and can be implemented in a more efficient way.

- Analysis to reveal why the prototype refinement, or prototype selector which might be more appropriate, works is missing. For example, the distribution of $W^*$ is not explcitly studied. If $W$ tends to be more one-hot, it validates the hypothesis that $W$ acts like a prototype selector.

- The proposed method requires solving the optimization problem in Eq (7) for every single query sample which introduces additional overhead.

**Questions:**

- More analysis and explanations on the optimal transport regularization is required.

- A thorough analysis into the prototype refinement is necessary. For example, the distribution of $W^*$ may reveal the true effectiveness of the ``prototype refinement'' module.

- Evaluate closed-form solution to $W$ in Eq (7).

---

> ### Author Response · Authors · 2024-11-22
>
> A1: The first term of dis() in Eq. 3 aims to refine prototypes in the original memory by transferring statistics of query features. Also as reviewer questioned that such kind of refinement seems much more like selection from the original memory, we have also discussed from line 208 to 210 that our proposed refinement belongs to composition refinement. As for the second term of OT()  in Eq. 3, instead of optimizing the distribution between prototypes and query features as reviewer understand, we aim to minimize the gap between the refined prototypes and prototypes in the original memory. That is to say, we want to ensure that the refined prototypes do not contain anomalous features. Otherwise, it should be harmful for IAD task. To sum up, the two terms are all necessary for robust IAD in low data regime.
>
> A2: (1) We agree with the understanding of reviewer that such kind of refinement seems much more like selection from the original memory, and we have also discussed from line 208 to 210 that our proposed refinement belongs to composition refinement.
> (2) We mainly take care of a memory efficient online IAD manner, thus, in our current implementation, the prototypes’ refinement can only be benefited from the current query features rather than the historical ones. However, our model could be directly adapted into such kind of historical aware online manner. Specifically, given the current query features $\boldsymbol{f}_t^q$, once the transform matrix $\boldsymbol{W}$ is well-optimized, we augment $\boldsymbol{W}\mathcal{\boldsymbol{M}}$ into another space independent with original memory $\mathcal{\boldsymbol{M}}$. When the next query features coming, we use the original memeory and a downsampled augmented memory for the next round optimization and anomaly detection. We report results of our developed PatchCore+ and the reviewer suggested historical awared PatchCore+ (called Online PatchCore+) in Table 6 of the appendix. For convenience, we show the Table 6 of the appendix as follows:
>
> | Method              | 1-shot Image | 1-shot Pixel | 2-shot Image | 2-shot Pixel |
> |:--------------------|:------------:|:------------:|:------------:|:------------:|
> | PatchCore           | 71.0         | 96.3         | 71.4         | 96.5         |
> | PatchCore+          | 74.9         | 96.6         | 78.2         | 96.9         |
> | Online PatchCore+   | **75.2**     | **97.1**     | **78.5**     | **97.2**     |
>
> A3: Taking Euclidean distance for example, we replenish the closed form solution on optimizing W by minimizing the objective in Eq. 7 w.r.t. the terms associated with W and obtain the closed solution as follows:
>
> $\mathcal{\boldsymbol{W}}=(\boldsymbol{f}_t^q \mathcal{\boldsymbol{M}}^T+\boldsymbol{T}  \mathcal{\boldsymbol{M}}  \mathcal{\boldsymbol{M}}^T)(\mathcal{\boldsymbol{M}}  \mathcal{\boldsymbol{M}}^T)^{-1}/(1+\frac{\lambda}{n} \boldsymbol{T} \textbf{1})$
>
> where $\boldsymbol{T}$ is transport matrix, $\textbf{1}$ is a all one vector.
> There are two important observations for implementation: 1) $\boldsymbol{W}$ could be updated in parallel, thus it’s efficient; 2) the inversion of $\mathcal{\boldsymbol{M}}  \mathcal{\boldsymbol{M}}^T$ is time-consuming since the dimensionality is high. Fortunately, the inversion itself is independent with updating $\boldsymbol{W}$, we can compute it in advance. Hence, the inference speed could be never bothered. Additionally, we further compare PatchCore, PatchCore+ ($\boldsymbol{W}$ is optimized by SGD) and Closed PatchCore+ ($\boldsymbol{W}$ is optimized by closed form) and report results in Table 7 of the appendix. The results reveal that the closed form solution on $\boldsymbol{W}$ is more efficient than SGD, and we will also update the closed form solution in our final manuscript. For convenience, we report the Table 7 of the appendix as follows:
>
> | Method              | Image AUROC | Pixel AUROC | Inference Time (s) |
> |:--------------------|:-----------:|:-----------:|:------------------:|
> | PatchCore           | 71.4        | 96.5        | 0.20              |
> | PatchCore+          | 78.2        | 96.9        | 0.50              |
> | Closed PatchCore+   | 78.0        | 97.0        | 0.36              |

---

> ### Author Response · Authors · 2024-11-22
>
> A4: We replenish visualizations on the distribution of $\boldsymbol{W}^*$ in Fig. 7 of the appendix. Each line represents the refinement coefficients for selecting from prototypes in original memory, which we find is also sparse. Furthermore, we take features of foreground, background and anomaly into consideration and visualize the corresponding selected items in Fig. 7 (c), with the highest Top-3 coefficients in $\boldsymbol{W}^*$. As we can see that, the foreground and background tend to use prototypes representing foreground and background, respectively. Moreover, anomaly tends to use foreground prototypes in original memory, which also consistent with our intuition. We will also update the corresponding analysis on $\boldsymbol{W}^*$ in the final manuscript.
>
> A5: Our model is a specific case of test-time training (TTT) in anomaly detection, TTT is one of the most popular concept recently [1]. Although it introduces extra time-consuming at inference, it is indeed helpful for aligning test scenarios with the training ones, thus effectively mitigating overfitting. What is important is that we should strike some good balance between additional overhead and promising performance. We have reported the performance and inference time in Table 3 of the manuscript. Additionally, as reviewer suggested, we also replenish the performance of our model using closed form solution to optimize $\boldsymbol{W}$ in Table 7 of the appendix. Both of them verify that our model could achieve balance between performance and real-time efficiency.
>
> [1] Test-Time Training with Self-Supervision for Generalization under Distribution Shifts.

---

> > ### Author Response · Authors · 2024-11-23
> >
> > Dear Reviewer MiAm,
> >
> > We appreciate it if you could let us know whether our responses are able to address your concerns. We're happy to address any further concerns. Thank you,
> >
> > Best wishes.
> >
> > Paper 5785 Authors

---

> > > ### Author Response · Authors · 2024-11-25
> > >
> > > Dear Reviewer MiAm,
> > >
> > > Given that the rebuttal phase is about to end, we would like to know if our responses have addressed your concerns. Additionally, we are still here and happy to address any further concerns.
> > >
> > > Best wishes,
> > >
> > > Paper 5785 Authors

---

> > > ### Comment · Reviewer_MiAm · 2024-11-26
> > >
> > > Thanks for the discussion on the difference between prototype refinement and optimal transport. However, a few questions still remain unresolved.
> > >
> > > - First, the OT does not necessarily suppress the anomalies in query sample. For example, if there are anomalous in $p_s$ the anomalies are still assigned to multiple normal prototypes in $q_s$. This is due to the constraint applied to solving the OT problem, i.e. $\sum_i^n T_{i,j}=1/m$.
> > >
> > > - The newly added Fig 7 is hard tp comprehend. It is unclear whether the transformation matrix is acting like a selection.
> > >
> > > - Test-time training, to the best of my knowledge, aims to address the distribution shift between training and testing datasets. Since there is no distribution between training and testing under the few-shot anomaly detection scenario, I feel it might be more appropriate to evaluate the method under testing data subject to distribution shift.
> > >
> > > - More importantly, if this paper aims to address training anomaly detection model at test-time, existing methods dealing with noisy training dataset, i.e. training data contains anomalies, should be compared [1]. These works can be used to build an anomaly detection model from scratch or few-shot pre-trained model on the testing data stream.
> > >
> > > [1] Chen, Yuanhong, et al. "Deep one-class classification via interpolated gaussian descriptor." Proceedings of the AAAI Conference on Artificial Intelligence. Vol. 36. No. 1. 2022.
> > >
> > > Overall, I feel the submission should be improved in better explaining the necessity of optimal transport. If test-time training is the main focus, evaluation on distribution shifted data might be more convincing. Alternatively, comparison with anomaly detection methods that can train a model from noisy data, i.e. contaminated with anomalies, is necessary.

---

> ### Author Response · Authors · 2024-11-26
>
> Dear reviewer MiAm,
>
> Thanks for your reply. We gonna response to your remaining concerns from the following aspects:
>
> (1) The goal for OT in our task that suppressing anomalies by aligning distribution between refined prototypes and original prototypes can be expressed as minimizing the following objective:
>
> $\mathcal{L} = \sum_{i,j} \boldsymbol{T}(i,j) \||\boldsymbol{W}(i,:) \mathcal{\boldsymbol{M}}_s - \mathcal{\boldsymbol{M}}_s(j,:)\||_2$
>
> where $\sum_{i} \boldsymbol{T}(i,j)=\frac{1}{m}$ and $\sum_{j} \boldsymbol{T}(i,j)=\frac{1}{n}$. That is $p_s$ is expanded by $\boldsymbol{W} \mathcal{\boldsymbol{M}}_s$ and $q_s$ is expanded by $\mathcal{\boldsymbol{M}}_s$.
>
> Let us consider a case that $p_s$ contains anomaly features. With the constraint of $\sum_{i} \boldsymbol{T}(i,j)=\frac{1}{m}$, although transport matrix may pay attention to the anomaly features in $p_s$, in order to minimize the term of OT objective above, in the next round of optimization on $\boldsymbol{W}$ (since we update $\boldsymbol{W}$ and $\boldsymbol{T}$ iteratively), theoretically, it may ignore those anomaly features by setting them with some small weights. Therefore, the anomalies could be suppressed.
>
> On the contrary, if we only adopt point-to-point distance regularization between $\boldsymbol{f}_t^q$ and $\boldsymbol{W} \mathcal{\boldsymbol{M}}_s$ as the first term in Eq. 3 without OT objective. Then the refined prototypes may be misled by anomalies in $\boldsymbol{f}_t^q$ so that the distance between $\boldsymbol{f}_t^q$ and $\boldsymbol{W} \mathcal{\boldsymbol{M}}_s$ could be smaller. Namely, the anomalies could not be removed without OT objective.
>
> To sum up, we claim that OT objective in Eq. 3 could suppress potential anomalies conveyed in the refined prototypes.
>
> (2) We replenish another $\boldsymbol{W}$ visualized on bottle (the bottle image is illustrated in Fig. 8 (a) of the appendix) with colorbar in Fig. 8 (b) of the appendix for better view. $\boldsymbol{W} \in \mathbb{R}^{1024 \times 102}$ in this example, where 1024 is the number of query features given the input bottle image, 102 is the number of prototypes in the original memory. As we can see that each row of $\boldsymbol{W}$ is sparse, the value of which represents the strength of current query feature selecting a specific prototype in the original memory. Additionally, we further visualize $\boldsymbol{W}$ corresponding to three different regions on the bottle image marked with red box including foreground region, background region, and anomaly region, results are reported in Fig. 8 (c) of the appendix. Taking foreground region for example, we denote the corresponding weight vector as $\boldsymbol{W}_{f} \in \mathbb{R}^{102}$.
> We select the highest Top-3 values in $\boldsymbol{W}_f$ and use their indexes to select the corresponding prototypes in the original memory. Then we visualize these selected prototypes using PCA on a 2-D space. For better view, we draw all prototypes in the original memory on the 2-D space and use different color to indicate prototypes that belonging to foreground or background. As we can see that, for query feature of foreground region, it also tends to select foreground prototypes. Similar analysis could be conducted on background region and anomaly region. Interestingly, we find that query feature of anomaly region also prefers to select foreground prototypes, which is consistent with our intuition as well.
>
> (3, 4) From the perspective of limitation in few-shot IAD, we feel that the main obstacle lies in overfitting when the normal training images are few, which is also a kind of distribuiton gap between the original prototypes and the current query features. Hence, our proposed prototype refinement could mitigate such gap in some extent by transferring statistics of the current query features (the first term in Eq. 3) while suppressing the potential anomalies in the current query features (the second term in Eq. 3). When we take original memory as training dataset and take query features as test dataset, it might be similar with TTT we guess. However, we mainly want to say that the spirit of reducing the gap between prototypes in the original memory and query features is a kind of similar with TTT. If there has any confusions, we are glad to modify them and make them more precise.

---

> > ### Comment · Reviewer_MiAm · 2024-11-26
> >
> > Thanks for explaining why OT could help alleviate the risk of associating anomalies with prototypes. But I am now more confused by the notations. If $p_s$ contains anomalies, does it mean the prototypes could be contaminated with anomalies? More specifically, in the case of PatchCore, does the few-shot support set contain anomalous samples?

---

> ### Author Response · Authors · 2024-11-26
>
> Dear reviewer MiAm,
>
> Thanks again for your reply and question. In order to keep consistency with our discussions before, we express $p_s$ and $q_s$ used in calculating OT as follows:
>
> $p_s=\sum_{j=1}^m \frac{1}{m} \delta_{\boldsymbol{W} \mathcal{\boldsymbol{M}}_s}$
>
> $q_s=\sum_{i=1}^n \frac{1}{n} \delta_{\mathcal{\boldsymbol{M}}_s}$
>
> Although the meanings of $p_s$ and $q_s$ are reversed compared with Eq. 4 in the manuscript, they are the same for optimization. As we can see that samples from $p_s$ (also called refined prototypes) may contain anomalies at very begining because of under-optimization while samples from $q_s$ (also called original prototypes) must be normal since $\mathcal{\boldsymbol{M}}_s$ is fixed and it only contains normal features as in PatchCore does. By fixing original prototypes, on the one hand, we could ensure to avoid contamination $\mathcal{\boldsymbol{M}}_s$ with anomalies. On the other hand, we could also provide correct guidance for suppressing potential anomalies in the refined prototypes ${\boldsymbol{W} \mathcal{\boldsymbol{M}}_s}$.
>
> We hope that our response could well address your confusion. And we will strength corresponding description in Sec. 4.1 to make it clearer in the final version of manuscript.
>
> We're happy to address any further concerns. Thank you,
>
> Best wishes,
>
> Paper 5785 Authors

---

> > ### Author Response · Authors · 2024-11-28
> >
> > Dear reviewer MiAm,
> >
> > We appreciate it if you could let us know whether our responses are able to address your concerns. We're happy to address any further concerns. Thank you,
> >
> > Best wishes.
> >
> > Paper 5785 Authors

---

> > ### Author Response · Authors · 2024-11-30
> >
> > Dear Reviewer MiAm,
> >
> > Given that the rebuttal phase is about to end, we would like to know if our responses have addressed your concerns. Additionally, we are still here and happy to address any further concerns.
> >
> > Best wishes,
> >
> > Paper 5785 Authors

---

> > ### Comment · Reviewer_MiAm · 2024-12-02
> >
> > Thanks for your further clarification. However, I am not fully convinced that $p_s$, as in its current form $p_s=\sum_{j=1}^m \frac{1}{m}\delta_{\mathbf{W}\mathcal{M}_s}$, could contain significant anomalies. Because the author revealed that the transformation $\mathbf{W}$ acts more like a selection from $\mathcal{M}_s$. It unlikely that simple linear combination of $\mathcal{M}_s$ could represent any unseen anomalies. There is not enough evidence to support the claims either. I think the authors should provide more in-depth analysis into the effectiveness of OT regularization, e.g. by provide more experimental evidence or convincing analysis.

---

> ### Author Response · Authors · 2024-12-02
>
> Thanks for your further questions.
>
> (1) According to Eq. 3, if we only employ the first term, the objective would be minimal when $\boldsymbol{f}_t^q=\boldsymbol{W}^* \mathcal{\boldsymbol{M}}_s$. Although $ \mathcal{\boldsymbol{M}}_s$ contains normal features, there is still possible for the model to reconstruct normal features similar with anomalies.
>
> (2) To verify the claims above, we have made comprehensive experiments in ablation studies of Table 5 and Fig. 4, where $\boldsymbol{W}^*$ and $\boldsymbol{T}^*$ separately represents employing the first term and the second term in Eq. 3. As we can see that using $\boldsymbol{W}^*$ can improve performance thanks to transferring statistics of the current query features. On the other hand, the performance could be further enhanced by employing $\boldsymbol{T}^*$, which should be attributed to the fact that suppressing potential anomalies in $\boldsymbol{W}^* \mathcal{\boldsymbol{M}}_s$ is also crucial for few-shot IAD.
> For convinence, we report Table 5 as follows:
>
> ** Table 5:** Ablation studies of WinCLIP+ with image/pixel-level AUROCs under 2 shot. The best results are in bold.
>
> | $W^*$       | $T^*$       | MVTec Image | MVTec Pixel | VisA Image | VisA Pixel | MPDD Image | MPDD Pixel |
> |:-------------:|:-------------:|:-----------:|:-----------:|:----------:|:----------:|:----------:|:----------:|
> | **$\times$**  | **$\times$**  | 93.7        | 93.8        | 83.8       | 95.1       | 72.5       | 96.5       |
> | **$\checkmark$** | **$\times$** | 93.7        | 94.7        | 83.7       | 96.2       | 74.9       | 96.4       |
> | **$\checkmark$** | **$\checkmark$** | **93.9**   | **96.2**   | **84.1**  | **96.4**  | **76.0**  | **97.3**  |
>
> We hope that our response could well address your confusion.
>
> We're happy to address any further concerns. Thank you,
>
> Best wishes,
>
> Paper 5785 Authors

---

> ### Author Response · Authors · 2024-12-02
>
> Dear Reviewer MiAm,
>
> We sincerely appreciate your insightful comments and the acknowledgment of our paper from the closed form optimization and presentation sides. As the rebuttal deadline approaches, we would like to know whether there are any additional concerns we can address. We will try our best to provide further clarification.
>
> Few-shot IAD is a promising but challenging setting where most current works try to avoid it resulting in limited generalizability. The proposed prototype refinement method is the first work managing to provide a unified view for solving memory-based (or say prototype-oriented) few-shot IAD efficiently and effectively, which is also a flexible method for solving other prototype-oriented tasks like video anomaly detection and prediction (Please refer to response for reviewer Exqa). None of the above-mentioned will be possible without your support.
>
> We hope our provided response during the rebuttal addresses the reviewers' concerns and the reviewers can re-evaluate our paper. We thank the reviewers once again for their efforts.
>
> Best regards,
> Authors of Paper 5785

---

### Author Response · Authors · 2024-11-22

Dear reviewers and AC

Thanks a lot for your effort in reviewing this submission! We have tried our best to address the mentioned concerns/problems in the rebuttal. Feel free to let us know if there is anything unclear or so. We are happy to clarify them.

Best, Authors

---

### Author Response · Authors · 2024-11-25

Dear AC,

Given that the rebuttal phase is about to end, however, it seems that reviewer MiAm hasn't read our responses. Reviewer MiAm's suggestions really do help us to improve the quality of this work, and we wonder that whether our responses addressed their concerns.  Could you please remind the reviewer to check out our responses.

Best wishes.

Paper 5785 Authors

---

### Comment · Area_Chair_ZaTu · 2024-12-01

Hi Reviewers,

The authors have provided new results and responses - do have a look and engage with them in a discussion to clarify any remaining issues as the discussion period is coming to a close in less than a day (2nd Dec AoE for reviewer responses).

Thanks for your service to ICLR 2025.

Best,
AC

---

### Meta-Review · Area_Chair_ZaTu · 2024-12-22

**Metareview:**

This paper introduces a test-time training algorithm for memory-based anomaly detection (AD) methods to address the few-shot anomaly detection problem in industrial contexts. It adapts the prototypes used for a new query as a linear combination of existing prototypes while including an optimal transport based regularizer to suppress anomalies in query images. An alternating minimization approach is used to solve for prototype updates then for optimal transport. The method is evaluated on 4 relevant industrial anomaly detection datasets where it shows improvements when applied onto recent memory-based AD methods.

Key strengths of the paper are that it addresses a practically relevant problem of few-shot industrial anomaly detection with a somewhat novel test-time training approach and that it is applicable to the class of memory-based AD methods which are state-of-the-art. Evaluations show improvements on relevant industrial AD datasets.

Key weaknesses are that the optimization objective proposed, in particular the use of optimal transport, is not well motivated. The paper claims it is used to suppress anomalies but there is insufficient analysis to support this claim. There is also a lack of analysis of the results given that improvements are uneven across the different datasets and are marginal in several cases.

Overall this paper presents a novel an interesting approach to an important problem. However, the claims and motivation around the optimal transport component, a key part of the framework, are not well justified, and the paper lacks a detailed analysis of the variation in results. As such, the AC recommends rejection for this borderline paper.

**Additional Comments On Reviewer Discussion:**

Key concerns raised were lack of sufficient motivation and justification of the optimal transport component, variability in the performance improvements (that were sometimes marginal) and lack of comparison to recent baselines. The authors provided additional experimental results on a recently released dataset (Real-IAD) and comparisons with additional methods (GraphCore, FOCT) that addressed the relevant concerns. While there was a discussion and clarifications about the motivation and justification of the optimal transport component, reviewer MiAm remained unconvinced. Reviewers were also not convinced by the responses to the requests for additional analysis of experiment results.

Overall, while the additional experiments showed promising results, the AC agrees that further justification and analysis of the method is required given that OT is a key component and the high variability in the results (in several cases marginal improvements are seen), as discussed above.

---

### Decision · Program_Chairs · 2025-01-22

Reject